# Why are Visually-Grounded Language Models Bad at Image Classification?

**Yuhui Zhang**[1][†]    **Alyssa Unell**[1]    **Xiaohan Wang**[1]    **Dhruba Ghosh**[2]    **Yuchang Su**[3]
**Ludwig Schmidt**[1,2][†]    **Serena Yeung-Levy**[1][†]
[1]Stanford University    [2]University of Washington    [3]Tsinghua University
[†]{yuhuiz,ludwigsc,syyeung}@stanford.edu

## Abstract

Image classification is one of the most fundamental capabilities of machine vision intelligence. In this work, we revisit the image classification task using visually-grounded language models (VLMs) such as GPT-4V and LLaVA. We find that existing proprietary and public VLMs, despite often using CLIP as a vision encoder and having many more parameters, significantly underperform CLIP on standard image classification benchmarks like ImageNet. To understand the reason, we explore several hypotheses concerning the inference algorithms, training objectives, and data processing in VLMs. Our analysis reveals that the primary cause is data-related: critical information for image classification is encoded in the VLM's latent space but can only be effectively decoded with enough training data. Specifically, there is a strong correlation between the frequency of class exposure during VLM training and instruction-tuning and the VLM's performance in those classes; when trained with sufficient data, VLMs can match the accuracy of state-of-the-art classification models. Based on these findings, we enhance a VLM by integrating classification-focused datasets into its training, and demonstrate that the enhanced classification performance of the VLM transfers to its general capabilities, resulting in an improvement of 11.8% on the newly collected ImageWikiQA dataset.[1]

## 1   Introduction

The ability to recognize objects within images is a fundamental capability of machine vision. Over the past 15 years, the field has experienced significant breakthroughs due to deep learning and large-scale datasets [11, 48]. For instance, on the renowned ImageNet dataset, designed to classify images into 1,000 categories, the error rate has dramatically decreased from 47.1% in 2009 to 9.1% in 2024, representing a 5-fold reduction [30, 12]. Consequently, these classification models have superseded most human labelers.

Nowadays, the community has focused on more sophisticated and nuanced capabilities in the quest for visual intelligence. Visually-grounded language models (VLMs), which integrate visual signals from vision encoders with large language models, have recently emerged as a promising paradigm [2, 36, 32]. VLMs like GPT-4V [36], Gemini-1.5 [44], or Claude-3 [3] have demonstrated advanced visual understanding abilities, such as answering math questions from table images or generating HTML code from design sketches.

In this work, we revisit the fundamental task of image classification using VLMs. Surprisingly, we find that various public and proprietary VLMs struggle with image classification in both open-world settings, where the class list is unknown, and closed-world settings, where class names are provided in the context (§2). Despite having many more parameters, there is a significant gap between the

---

[1]Project page: `https://yuhui-zh15.github.io/VLMClassifier-Website/`.

38th Conference on Neural Information Processing Systems (NeurIPS 2024).

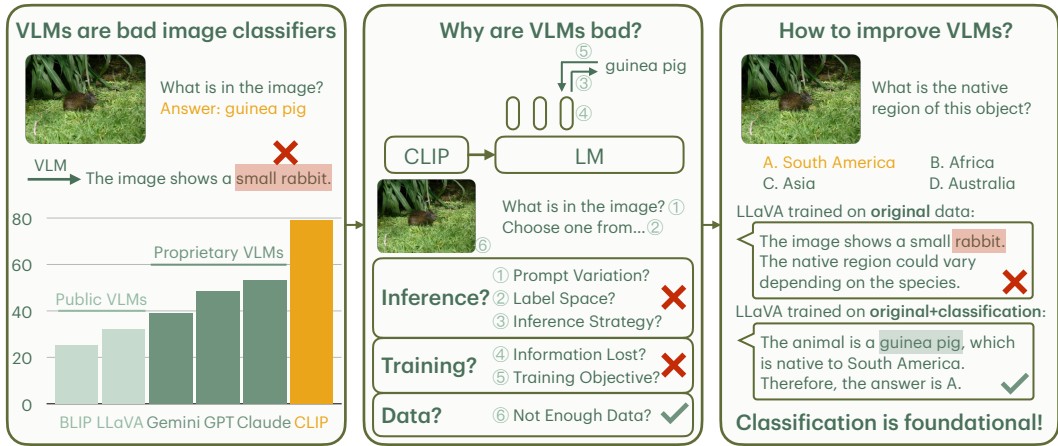

Figure 1: **Overview.** *(Left)* Different visually-grounded language models (VLMs) underperform CLIP in classification by a large margin, though they often use CLIP as a vision encoder. *(Middle)* We investigate several hypotheses about why VLMs are bad classifiers and find that the main reason is data. Critical information for image classification is encoded in the VLM's latent space but can only be decoded with enough data during VLM training. *(Right)* Based on our analysis, we improve a VLM by integrating classification data into its training, and find that the improved classification capabilities serve as foundations for more advanced capabilities such as visual question answering.

performance of VLMs and their commonly used vision encoder CLIP [37]. Our evaluation protocol involves feeding each image and a list of class names (in the closed-world setting) to the VLM as context and asking what is in the image; success is defined by whether the generated output contains the ground-truth class name.

To understand why VLMs underperform in classification settings, we investigate several hypotheses regarding VLMs' inference (such as prompt variations, label set size, inference strategy; §3.1), training (such as information lost, training objective; §3.2), and data (such as data-performance correlation; §3.3). Our extensive analyses suggest that the primary reason for the observed gap is data. We find that the information necessary for classification is encoded in the VLM's latent space but can only be decoded with proper training data. Specifically, there is a strong correlation between class presence during VLM training and performance in those classes. Furthermore, training VLMs on classification datasets achieves the same performance level as state-of-the-art classification models.

Motivated by our analysis, we propose a simple method to enhance VLMs' general capabilities by integrating traditional classification-focused datasets into VLM training (§4). We believe that classification is the foundation for more complex, advanced visual capabilities; for example, recognizing an object is a prerequisite for answering complex questions about it. To verify this, we created ImageWikiQA, which contains complex real-world questions about ImageNet objects. On ImageWiki-iQA, we find that VLMs fine-tuned on the ImageNet classification dataset achieve substantially higher accuracy in recognizing these objects and provide more accurate answers to these non-classification questions, outperforming pre-trained VLMs by 11.8%. This suggests that classical classification data can be beneficially reused in the VLM training process to enhance VLM performance.

In summary, our contributions are threefold:

- **Evaluating VLM classification weaknesses:** Our evaluations using ten VLMs across four benchmarks show that VLMs significantly lag behind CLIP in classification, uncovering a gap unaddressed by previous research.

- **Analyzing causes of poor classification performance:** Testing various hypotheses, we find that the lack of alignment data, rather than information lost or training objective, is the primary reason for VLMs' underperformance in classification.

- **Enhancing VLMs with classification data:** By adding classification data, we improve VLMs' accuracy in object recognition and overall performance, supporting that classification capability is foundational for complex object-related reasoning.

| Model | Open-World Setting | | | | Closed-World Setting | | | |
|---|---|---|---|---|---|---|---|---|
| | 🎯 | 🌻 | 🚗 | Ⓒ | 🎯 | 🌻 | 🚗 | Ⓒ |
| **Public VLM** | | | | | | | | |
| `BLIP2-2.7B` [27] | 25.3 | **27.0** | 0.0 | 46.9 | N/A | 14.2 | 2.7 | 22.3 |
| `IBLIP-7B` [10] | 14.6 | 1.9 | 0.0 | 36.5 | N/A | **26.8** | N/A | 58.4 |
| `IBLIP-13B` [10] | 14.7 | 2.4 | 0.0 | 36.4 | N/A | 20.0 | N/A | 59.5 |
| `LLaVA1.5-7B` [32] | 22.8 | 5.9 | 0.0 | 47.1 | N/A | 10.2 | 0.0 | 62.1 |
| `LLaVANeXT-V7B` [32] | 29.4 | 12.8 | 0.0 | 52.5 | N/A | 8.5 | 0.0 | 66.6 |
| `LLaVA1.5-13B` [32] | 24.3 | 5.3 | 0.0 | 49.9 | N/A | 7.2 | 0.1 | 70.9 |
| `LLaVANeXT-M7B` [32] | **32.3** | 17.7 | 0.0 | **54.2** | N/A | 16.1 | **3.6** | **77.3** |
| **Proprietary VLM** | | | | | | | | |
| `Claude3` [3] | **53.6** | **51.2** | **0.3** | **68.6** | 51.1 | 58.3 | 45.1 | 90.9 |
| `GeminiPro` [44] | 39.2 | 10.5 | 0.1 | 60.1 | 56.0 | 62.0 | **66.6** | 91.6 |
| `GPT4` [36] | 48.5 | 51.0 | 0.1 | 61.0 | **60.6** | **79.9** | 58.2 | **94.2** |
| **CLIP** | | | | | | | | |
| `CLIP-L` [37] | N/A | N/A | N/A | N/A | 74.8 | 76.0 | 77.5 | 95.8 |
| `EVA-G` [43] | N/A | N/A | N/A | N/A | **79.2** | **81.0** | **90.2** | **97.9** |

Table 1: **Evaluations of VLMs and CLIPs on standard image classification benchmarks.** VLMs exhibit poor performance in image classification, significantly lagging behind CLIP models. 🎯=ImageNet [11], 🌻=Flowers102 [35], 🚗=StanfordCars [21], Ⓒ=Caltech101 [13].

## 2 VLMs are Bad at Image Classification

We begin by evaluating state-of-the-art visually-grounded language models (VLMs) using standard image classification benchmarks. Our findings reveal that these VLMs significantly underperform compared to state-of-the-art classification models, such as CLIP.

### 2.1 Models

**VLMs.** We selected ten widely-used state-of-the-art VLMs, covering different architectures, training methods, and data. These VLMs include three proprietary ones, `GPT-4-Turbo` (shortened as GPT4, same below) [36], `Gemini-Pro-Vision` (GeminiPro) [44], and `Claude-3-Opus` (Claude3) [3], and seven public ones, `LLaVA1.5-Vicuna7B/13B` (LLaVA1.5-7/13B) [32], `LLaVANeXT-Mistral7B/Vicuna7B` (LLaVANeXT-M7B/V7B) [31], `BLIP2-OPT2.7B` (BLIP2-2.7B) [27], and `InstructBLIP-Vicuna7B/13B` (IBLIP-7/13B) [10]). Details of these models are provided in Appendix §A.1.

**CLIPs.** For comparison, we used two state-of-the-art image classifiers, `CLIP-ViT-L/14-336px` (shortened as CLIP-L, same below) [37] and `EVA-ViT-G/14` (EVA-G) [43]. Notably, `CLIP-L` and `EVA-G` are utilized by the LLaVA series [32] and the BLIP series as vision encoders [27], respectively. Therefore, the VLMs should theoretically have the same classification capacity as these vision models. Details are listed in Appendix §A.1.

### 2.2 Data

We evaluated the aforementioned models on four widely-used image classification benchmarks: ImageNet (shortened as 🎯, same below) [11], Flowers102 (🌻) [35], StanfordCars (🚗) [21], and Caltech101 (Ⓒ) [13], which contain 50,000, 6,149, 8,041, and 4,331 test images from 1,000, 102, 196, and 101 classes, respectively. ImageNet and Caltech cover more coarse-grained objects, while Flowers and Cars cover more fine-grained objects. Further details are provided in Appendix §A.2.

### 2.3 Evaluation Protocol

**VLMs.** We performed image classification in two settings: an open-world setting where the label set is not provided and a closed-world setting where classes are concatenated in the prompt. We

feed the image and the prompt to the VLM and let the VLM complete the rest of the tokens. One closed-world example is: "<image> What type of object is in this photo? Choose one from <class name A>, <class name B>, ..." We define success on a single example as whether the ground-truth label is included in the VLM generation. We report the success rate of all test examples.

**CLIPs.** CLIP can only be used in a closed-world setting where the label set is known. Following Radford et al. [37], we used the prompt "a photo of a <class>" to generate the text feature. For each image, we selected the class with the highest cosine similarity to the image feature. We did not include prompt ensembling to fairly compare with VLM. We report the accuracy of all examples.

## 2.4 Results

Table 1 reports the performance of different VLMs and CLIP models on these classification datasets.

We find that **VLMs exhibit poor performance in image classification, significantly lagging behind CLIP models**. For instance, on the ImageNet dataset, the best proprietary VLM, GPT4, only achieves an accuracy of 60.6% in the closed-world setting, whereas the best CLIP, EVA-G, attains an accuracy of 79.2%. In the open-world setting, the best public VLM, LLaVANeXT-M7B, achieves just 32.3% accuracy. The performance disparity is even more pronounced in fine-grained classification datasets like Flowers102 and StanfordCars. Notably, all LLaVA models use CLIP-L as the vision encoder, and although the total parameter count of these models is at least 20 times greater than that of the vision encoder, they significantly underperform compared to it.

Moreover, we find that **closed-world setting often outperforms open-world setting**. This is expected as the provided label set narrows the prediction space. However, since closed-world settings require including all class names in the context, they can result in an extremely long context that leads to high costs or even exceeds the VLM's context limit. For example, most public VLMs only support 4K context length, which cannot feed 1K ImageNet classes. We also find that **larger and better LMs slightly improve VLM performance.** For instance, LLaVA1.5-13B outperforms LLaVA1.5-7B, and LLaVANeXT-M7B outperforms LLaVANeXT-V7B.

# 3 Why are VLMs Bad Image Classifiers?

Given that visually-grounded language models (VLMs) underperform CLIPs at classification by a large margin, as reported in §2, we seek to understand the reasons behind that. We investigate several hypotheses concerning major differences between VLMs and CLIPs, which can be generally categorized into inference (§3.1), training (§3.2), and data (§3.3):

1. We start with inference-related questions. For example, does prompt variation, such as chain of thought, affect final performance? Does reducing the label set size in context narrow the gap between VLMs and CLIPs? Does performing probabilistic inference to force the generation into the label set help? We find none of these factors can fully close the gap between VLMs and CLIPs.

2. Therefore, we switch to training-related questions. For example, is the visual information from the vision encoder still preserved in the VLM's latent space? Is the text generation objective as effective as cross-entropy loss for learning classification? Surprisingly, the results show that the information is preserved, and the text generation objective is adequate for learning classification.

3. Finally, we investigate data-related questions. For example, does the VLM training data include enough classification data and cover enough classes? We find a strong correlation between class exposure in training and model performance. Moreover, VLMs can achieve the same level of performance as CLIPs when trained with enough data. These results suggest that data is the primary cause and effective solution for the poor classification performance of VLMs.

## 3.1 Inference

In this section, we investigate three questions related to VLM's inference, including prompt variation, label set size, and inference strategy.

**Prompt variation.** It is well known that language models (LMs) are sensitive to prompts [7, 16, 53]. To understand the effect of prompts on classification performance, we tested three semantically

| Method | LLaVA1.5-7B | | | | BLIP2-2.7B | | | |
|---|---|---|---|---|---|---|---|---|
| | 🎯 | 🌻 | 🚙 | Ⓒ | 🎯 | 🌻 | 🚙 | Ⓒ |
| **Prompt Variation** | | | | | | | | |
| Base Prompt | **22.8** | 5.9 | 0.0 | 47.1 | 25.3 | 27.0 | 0.0 | 46.9 |
| w/ Prompt Alternative 1 | 19.7 | 3.5 | 0.0 | 45.4 | **27.6** | **38.8** | 0.2 | 47.8 |
| w/ Prompt Alternative 2 | 21.6 | 6.6 | 0.0 | 48.1 | 24.3 | 30.6 | 0.1 | 47.9 |
| w/ Label (Fixed Order) | N/A | 6.1 | 0.1 | **70.5** | N/A | 28.0 | 2.1 | **53.8** |
| w/ Label (Random Order) | N/A | 10.2 | 0.0 | 62.1 | N/A | 14.2 | **2.7** | 22.3 |
| w/ Label (Random) + CoT | N/A | **18.1** | **0.4** | 64.5 | N/A | 8.5 | 0.0 | 16.1 |
| **Inference Strategy** | | | | | | | | |
| Direct Generation | 22.8 | 5.9 | 0.0 | 47.1 | 25.3 | 27.0 | 0.0 | 46.9 |
| Prob Inference (Sum Tokens) | 34.8 | 14.5 | 26.7 | 77.8 | 21.0 | 34.8 | 48.8 | 36.8 |
| Prob Inference (Avg Tokens) | 35.3 | 16.5 | 18.2 | 65.6 | 5.1 | 19.9 | 1.2 | 12.3 |
| Prob Inference (Avg) w/ CFG | **47.6** | **26.8** | **48.8** | **85.6** | **38.7** | **54.1** | **69.2** | **70.3** |

Table 2: **Analysis of VLMs from the inference perspective.** *(Top)* We explore prompt variation such as wording, label order, chain-of-thought and find it has limited impact on the performance. *(Bottom)* We leverage the probabilistic inference strategy, which improves the performance but still fails to close the gap between VLMs and CLIPs. Results are from the official validation set.

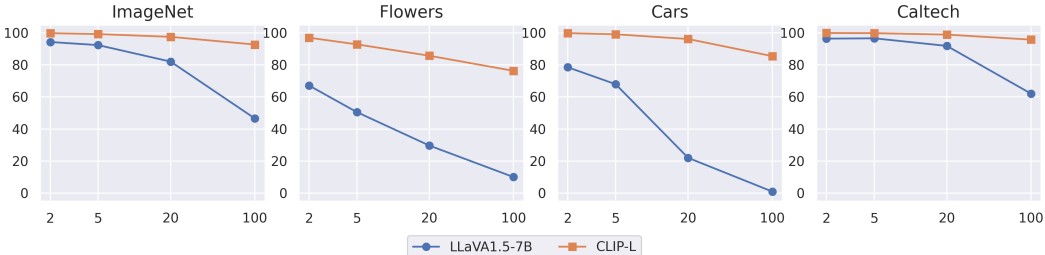

Figure 2: **Analysis of the label set size.** For each image, we randomly sample 100, 20, 5, 2 candidate classes from all the classes. The performance gap between VLMs and CLIPs becomes smaller when the number of classes is reduced. X-axis: number of classes; Y-axis: accuracy (%).

similar but differently worded prompts, compared feeding the label in dataset order or random order within the context, and leveraged the zero-shot chain-of-thought (CoT) prompting technique by adding "let's think step by step" at the end of the prompt [47, 20].

We find that **prompt variation has a limited impact on the performance.** From Table 2, we can see that changing the wording of prompts results in a performance variation within 3% for both `LLaVA1.5-7B` and `BLIP2-2.7B` on the ImageNet dataset. Different label orderings impact `LLaVA1.5-7B` less than `BLIP2-2.7B`. Chain-of-thought prompting consistently improves performance for instruction-tuned model `LLaVA1.5-7B` but not for `BLIP2-2.7B`.

**Label set size.** The label set size can be very large in practice (e.g., 1000 for ImageNet, 196 for StanfordCars), which results in an extremely long context when we concatenate all the class names. Since LMs often struggle with long contexts [33], we explore reducing the number of classes. For each image, we randomly select $K = 2, 5, 20, 100$ classes, always including the ground-truth label, and re-evaluate the VLM and CLIP performance.

We find that **the performance gap between VLMs and CLIPs narrows with reduced label size, but the gap always exists**. As shown in Figure 2, when evaluating `LLaVA1.5-7B` and `CLIP-L` on ImageNet, the gap decreases from 46% with 100 classes to 6% with 2 classes. However, the relative gap becomes larger, evidenced by a 23.9x error rate with 2 classes compared to a 7.3x error rate with 100 classes. The performance gap between VLMs and CLIPs always exists in all the settings.

**Inference algorithm.** The default inference algorithm for classification with VLMs is directly generating the class name given a prompt. As the generation is open-ended, even when provided with a list of candidate choices, the generation may not match one of the pre-defined classes. To mitigate this problem, we employ probabilistic inference techniques for VLMs. Specifically, for each class name,

| Model | Feature | 📊 | 🌻 | 🚗 | ⓒ |
|---|---|---|---|---|---|
| **Probing** | | | | | |
| LLaVA1.5-7B | Last Tok | 76.9 | 94.5 | 81.0 | 96.7 |
| LLaVA1.5-7B | Avg Tok | **77.1** | **96.2** | **82.8** | **97.3** |
| BLIP2-2.7B | Last Tok | 80.3 | 98.8 | 91.0 | 98.0 |
| BLIP2-2.7B | Avg Tok | **81.4** | **98.9** | **92.6** | **98.0** |

| Model | Trainable | 📊 | 🌻 | 🚗 | ⓒ |
|---|---|---|---|---|---|
| **Fine-tuning** | | | | | |
| CLIP-L | Linear | 85.2 | **98.6** | **91.5** | 97.6 |
| LLaVA1.5-7B | Proj | **85.7** | 97.6 | 90.4 | 97.5 |
| LLaVA1.5-7B | Proj+LM | NaN | 97.9 | 90.7 | **97.8** |
| EVA-G | Linear | 86.5 | **99.2** | **94.3** | 98.5 |
| BLIP2-2.7B | Proj | **88.0** | 99.0 | 93.9 | **98.8** |

Table 3: **Analysis of VLMs from the training perspective.** *(Left)* We conduct feature probing experiments on the VLM's last layer and find that the information required for classification is mostly preserved in the VLM's latent space. *(Right)* We fine-tune VLMs on the classification datasets using the text generation objective and find that the text generation training objective is as effective as the traditional cross-entropy for learning classification, which eliminates the VLM-CLIP performance gap, with VLMs now being the state-of-the-art classifier. Results are from the official validation set.

we compute prob(class name|image, prompt) and select the class name with the highest probability as the prediction. Since class names can consist of multiple tokens (e.g., "guinea pig" consists of two tokens), we either average the probabilities of all tokens or sum up all the tokens [7]. We also explore classifier-free guidance (CFG) techniques by ranking $t * \text{prob}(\text{class name}|\text{image}, \text{prompt}) + (1 - t) * \text{prob}(\text{class name}|\text{prompt})$ with varying guidance coefficients $t$ [38, 50].

We find that **the probabilistic inference method improves the performance, but the gap persists.** As shown in Table 2, LLaVA1.5-7B achieves 35.3% accuracy on ImageNet using probabilistic inference compared to 22.7% using the direct generation method. Adding classifier-free guidance further improves the performance, where LLaVA1.5-7B performance boosts to 47.6% on ImageNet. However, it still leaves around a 30% performance gap between VLMs and CLIPs, as its vision encoder CLIP-L achieves 74.8% on ImageNet. Moreover, the probabilistic inference approach is computationally expensive in practice because we need to compute the probability of each class.

### 3.2 Training

Since inference-based modifications fail to close the performance gap between VLMs and CLIPs, here we investigate two questions regarding to the training of VLMs. We study whether the visual information is lost in the VLM and whether the text generation objective is suitable for learning the classification task.

**Visual information lost in the VLM.** VLMs process images using an image encoder, such as CLIP, which has strong classification capabilities. We hypothesize that, during the propagation of image features output from the vision encoder in language model layers, the necessary information for classification is lost. To test this hypothesis, we conduct feature probing experiments. Specifically, the features from the VLM's last layer have a shape corresponding to the length of the inputs (image tokens plus text tokens using the open-world prompt). We take the average of these token features or the last token feature and train a simple linear classifier on top of these frozen feature representations. The linear classifier is trained on the training set and evaluated on the validation set. Training details are provided in Appendix B.4. A higher accuracy indicates that less information is lost.

Surprisingly, we find that **the information necessary for classification is largely preserved in the VLM's latent space; however, it cannot be effectively decoded**. In Table 3, we show that the probing accuracy of LLaVA1.5-7B on ImageNet is 77.1%, which is close to the 85.2% probing accuracy of CLIP-L, the vision encoder used by LLaVA1.5-7B. The same conclusion holds for BLIP2-2.7B and its vision encoder EVA ViT-G/14, demonstrating that most information is preserved during the VLM computational process. However, this information cannot be effectively decoded, as §2 shows that the best accuracy LLaVA1.5-7B can achieve on ImageNet is 47.6%.

**Training objective.** Since information is encoded in VLMs, we wonder whether we can train them to decode the information. VLMs can only be trained to perform classification by auto-regressively generating text-form labels. We hypothesize that this generative objective may be more difficult and less effective in learning classification compared to the traditional cross-entropy loss. To understand the effectiveness of this training objective, we explore fine-tuning VLMs on classification datasets. We

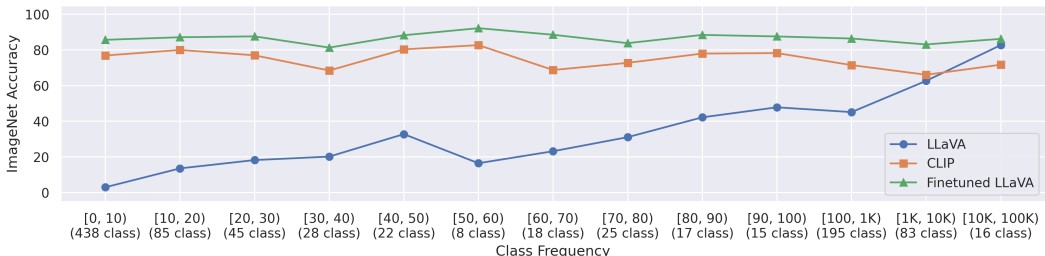

Figure 3: **Analysis of VLMs from the data perspective.** We study the relation between the ImageNet class frequency in the VLM training data and the VLM's classification performance on those classes. A strong correlation is observed, indicating that data determines VLM classification performance.

convert classification datasets into the text generation format using the template (e.g., "<image> What type of object is in this photo? <class name>"). We fine-tune two model architectures (`LLaVA1.5-7B` and `BLIP2-2.7B`) across different datasets and compare their accuracy to that of fine-tuned CLIP models. Additionally, we investigate fine-tuning different parts of the models, such as only fine-tuning the projector between vision encoder and language models or fine-tuning the projector along with the language models using LoRA [17]. Training details are provided in Appendix B.5.

We find that **the text generation training objective is as effective as the traditional cross-entropy for learning classification tasks, which eliminates the performance gap between VLMs and CLIPs, with VLMs now being the state-of-the-art classifier**. From Table 3, we find that `LLaVA1.5-7B` achieves 85.7% accuracy on ImageNet, same as 85.2% accuracy when fine-tuning its vision encoder `CLIP-L`. The same findings apply to all the VLMs on all the datasets.

Moreover, we find that **fine-tuning only the projector is sufficient and has better numerical stability**. From Table 3, we can see the same level of accuracy achieved by fine-tuning the projector and fine-tuning projector and LLM using LoRA on the three datasets. We also find that fine-tuning LLM with LoRA often leads to numerical instabilities, such as spikes in loss. While the loss sometimes returns to normal, other times it does not. For example, despite trying various hyperparameters for ImageNet, instability persisted (see Appendix §B.5). In contrast, fine-tuning only the projector always results in a steadily decreasing loss. Notably, the projector is often much smaller (e.g., a 2-layer MLP in LLaVA) compared to LLMs, suggesting the potential for parameter-efficient prefix tuning for VLMs [28].

## 3.3 Data

As observed, the visual information is preserved in the VLM's latent space, and the VLM's training objective is sufficient for learning classification tasks. We hypothesize that the poor classification performance of VLMs is due to their training data. For example, there might be insufficient classification data or a lack of diverse classes. To investigate this, we analyze the `LLaVA1.5-7B` training data [32], the only fully publicly available VLM training dataset. We examined the relationship between the frequency of class occurrence and the VLM's classification performance on those classes[2].

Our findings indicate that **data determines VLM classification performance**. As shown in Figure 3, there is a strong correlation between the presence of class labels in the VLM training data and the VLM's classification accuracy for those classes. The `LLaVA1.5-7B` model achieves 82.7% zero-shot accuracy on ImageNet classes with more than 10,000 occurrences in its training data, but only 3.0% accuracy on classes with fewer than 10 occurrences. The Spearman correlation between class frequency and class performance is very high (0.76 on ImageNet; see Appendix §B.6). In contrast, there is no correlation between its vision encoder `CLIP-L`'s performance on those classes with the same VLM training data, and after fine-tuning `LLaVA1.5-7B` on the ImageNet classification data, the strong correlation disappears. Combining all of our results, we conclude that data plays a critical role in determining the VLM's classification performance.

Given that data availability is critical for VLM classification performance, we now examine whether the data type specifically impacts results. For instance, to train a VLM to classify a class like "guinea pig," should it rely on classification-focused data (e.g., "<image> What type of object is in this photo?

---

[2]LLaVA training contains two stages. Here we combine the data from the two stages. More in Appendix B.6.

| Model | Fine-tuning Data Type | ☀ | 🚙 | C |
|---|---|---|---|---|
| LLaVA1.5-7B Zero-shot | - | 5.9 | 0.0 | 47.1 |
| CLIP-L Fine-tuned | Classification Data | 98.6 | 91.5 | 97.6 |
| LLaVA1.5-7B Fine-tuned | Classification Data | 97.6 | 90.4 | 97.5 |
| LLaVA1.5-7B Fine-tuned | Captioning Data | 92.0 | 85.4 | 95.7 |

Table 4: **Analysis of data types.** We fine-tune the VLM on the caption-focused data generated by GPT4 using the same experimental settings as Table 3 and find that **data is the main determining factor for VLM performance, and the data type does not matter much**.

Guinea pig") or can it use broader types, such as caption-focused data (e.g., "<image> Generate a caption for the image. A guinea pig playing in the grass") or VQA-focused data (e.g., "<image> What is the native region of this guinea pig? South America")?

In previous experiments (see §3.2), we fine-tuned the VLM with a classification-focused template. Here, we fine-tune the VLM on caption-focused data generated by prompting GPT4 to produce captions incorporating each class's ground-truth name. Specifically, we used the prompt: "Generate a short caption for the image. The caption must include the ground-truth label of the image, which is <class name>." This approach produces diverse captions per class, such as "A guinea pig playing in the grass" or "A close-up view of a guinea pig in an indoor environment." We then fine-tune the VLM on this caption-focused data using identical experimental settings and evaluate it against the model trained with classification-focused data. Success is determined by whether the generated caption for a validation image includes the ground-truth class name.

We find that **data presence is the primary factor for VLM classification performance, with data type playing a minimal role**. From Table 4, we observe that models fine-tuned on both classification- and caption-focused data perform similarly on Flowers102, StanfordCars, and Caltech101, achieving accuracy gains of at least 48.6% over the non-fine-tuned model. This suggests that enabling a VLM to classify a specific class does not require classification-specific data; any data containing the class name suffices.

**These findings emphasize a critical data-centric view of VLM training**. A common question for VLM training is whether alignment between vision encoder outputs and text generation necessitates multi-modal data with specific class labels, despite both vision and language components seeing these classes in uni-modal pre-training. Previous studies have suggested that this alignment stage may be data-efficient or even unnecessary [32, 18], likely due to their evaluation settings not requiring vision-centric fine-grained recognition. Our work, however, shows that 1) multi-modal data is essential for alignment and 2) increasing data quantity linearly boosts performance. Concurrent work from DeepMind supports our findings, showing that most VLM tasks benefit from pre-training on larger datasets [6].

# 4 Improving VLM with Classification Data

Based on the analysis presented in §3, in this section, we discuss the enhancement of visually-grounded language models (VLMs) by integrating classification-focused data into their training. We demonstrate that this data intervention not only boosts the VLM's classification performance but also enhances its general capabilities.

## 4.1 Motivation

Classification is fundamental to enabling more advanced capabilities of VLMs, such as visual question answering and reasoning. For example, suppose a virtual assistant is helping visually impaired individuals prepare mushrooms as food. In that case, the model must correctly identify the mushroom species to answer questions like "Is this mushroom poisonous?" The poor classification performance of VLMs lays a weak foundation for their advanced capabilities.

From §3, we identify the primary cause of poor classification performance is the lack of data. Therefore, we propose a straightforward solution: integrating classification-focused data into the VLM training process. We hope that incorporating classification data not only enhances classification accuracy but also improves general capabilities.

| Model | Acc | Model | Acc |
|---|---|---|---|
| **Naive Baselines** | | **Public VLM** | |
| Random | 25.0 | BLIP2-2.7B [27] | 21.7 |
| Max Freq | 25.9 | IBLIP-7B [10] | 36.3 |
| GPT4 w/ GT Class | 100.0 | LLaVANeXT-V7B [31] | 37.0 |
| GPT4 w/o Image | 0.0 | IBLIP-13B [10] | 37.5 |
| Human w/ GT Class+Wiki | 96.5 | LLaVA1.5-13B [32] | 37.8 |
| LLaVA1.5-7B w/ GT Class | 55.9 | LLaVA1.5-7B [32] | 38.0 |
| | | LLaVANeXT-M7B [31] | 41.9 |
| **Proprietary VLM** | | **Finetuned VLM** | |
| GeminiPro [44] | 49.1 | LLaVA1.5-7B Finetuned on ImageNet | 30.6 |
| Claude3 [3] | 54.3 | LLaVA1.5-7B Finetuned on ImageNet+LLaVA | **49.8** |
| GPT4 [36] | **61.2** | | |

Table 5: **Evaluations of VLMs on ImageWikiQA.** ImageWikiQA is a multiple-choice question-answering dataset collected by feeding the Wikipedia pages of ImageNet classes to GPT-4. We find that current VLMs perform poorly in answering these questions, suggesting that their poor classification performance is a fundamental limitation for more advanced capabilities. Integrating classification data into VLM training enhances both their classification and overall capabilities.

## 4.2 ImageWikiQA

To verify our hypothesis, we need a dataset that evaluates both the classification and advanced capabilities of VLMs. However, current visual question answering benchmarks, such as VQAv2 [15], MM-Vet [51], and MMMU [52], primarily focus on advanced capabilities, such as reasoning and knowledge grounding, rather than classification. The objects in these datasets are relatively simple, and questions can often be answered by identifying only the general category of an image, such as "mushroom" or "flower", rather than specific types of mushrooms or flowers. In contrast, classification datasets have no questions related to more advanced capabilities.

To bridge the gap between classification and advanced capabilities, we introduce ImageWikiQA, an object-centric, knowledge-intensive question answering dataset that combines both worlds. Each question in ImageWikiQA is a multiple-choice question with four options and one correct answer (e.g., "<image showing a guinea pig> What is the native region of this object? A. South America; B. Africa; C. Asia; D. Australia"). Although each question does not directly ask for the category of the object within the image, the question can only be accurately answered if the class of the object is correctly identified. Example questions from ImageWikiQA can be found in Figure 1 (right) and Appendix §C.1.

ImageWikiQA is created by generating questions based on Wikipedia pages of ImageNet classes using GPT4. Specifically, for each class in ImageNet, we parsed the Wikipedia content of the class following Bujwid et al. [8], and then provided this content along with a prompt to GPT4 to generate five questions per class. We instructed GPT4 to replace the class name with "this object" in the questions so that the ground-truth class name is not provided. We retained all questions that GPT4 could answer correctly with the ground-truth class name and could not guess the answer without the class name. To ensure the question quality generated by GPT4, four human annotators attempted to answer the questions with the ground-truth class name and Wikipedia page and achieved an accuracy of 96.5%. Afterward, we randomly sampled at most 3 ImageNet images for each question, rebalanced the choice distribution, and composed the final ImageWikiQA dataset. In total, there are 2000 multiple-choice questions, each with an image, question, four candidate choices, and a reference to Wikipedia sentences, with a random guess accuracy of 25.0% and max frequency accuracy of 25.9%.

## 4.3 Results

Table 5 presents the performance of various VLMs on the ImageWikiQA dataset.

We find that **current state-of-the-art VLMs perform poorly on answering these questions given images**. For example, GPT4 achieves 100% accuracy when the ground-truth class name is provided, but only achieves 61.2% accuracy with images. Similarly, Claude3 and GeminiPro only achieve 54.3% and 49.1% accuracy, respectively. These results indicate that the poor classification performance of VLMs is a fundamental limitation for more advanced capabilities.

Furthermore, we find that **integrating classification data into VLM training improves its classification and general capabilities**. We fine-tuned `LLaVA1.5-7B` on the ImageNet 1.28M classification data and original 665K LLaVA instruction-tuning data (training detail in §C.2), which is able to achieve 84.4% accuracy on ImageNet classification compared to 22.8% for non-fine-tuned models. This improvement in classification translates to an 11.8% accuracy boost on ImageWikiQA, demonstrating classification is indeed a foundation for VLM's advanced capabilities. However, it is worth noting that fine-tuning solely on the classification task harms general capabilities. When fine-tuning `LLaVA1.5-7B` on ImageNet only, their accuracy on ImageWikiQA drops to 30.6%. Therefore, fine-tuning should be performed on a combined dataset. Developing fine-tuning methods that prevent such catastrophic forgetting is a promising future research direction.

## 5 Related Work

**Visually-grounded language models.** Visually-grounded language models (VLMs) refer to a large family of models that integrate visual signals into language models by modeling $p(y_t|y_{<t}, x)$, where $y_i$ is a text token and $x$ is a visual input such as an image or video. Recently, many powerful VLMs have been developed, including proprietary ones like GPT-4V [36], Gemini [44], Claude [3], Flamingo [2] and public ones like LLaVA [32, 31], BLIP [27, 10], OpenFlamingo [4], Otter [26], Fuyu [1], QwenVL [5]. The typical architecture of VLMs comprises three components: a visual encoder (often CLIP [37]), a language model, and a projector that bridges the visual outputs and language model inputs. The projector can be as simple as a linear layer or MLP (e.g., LLaVA [32]) or a complex architecture such as Transformer with cross-modal attention (e.g., BLIP [27]). VLMs are usually trained on image-text captioning data [40, 39] and carefully designed instruction-tuning data [32], with both the vision encoder and language model initialized from pre-trained versions. In this work, we evaluate widely used VLMs in classification settings and analyze two prominent architectures, LLaVA [32] and BLIP [27].

**Analysis of visually-grounded language models.** While VLMs have demonstrated impressive performance, many questions remain unanswered. Recent works have explored their limitations from various perspectives, including architectures [34], training recipes [18, 24], data [14, 46], vision encoders [45], language models [2, 44], input resolution [41], and proposed solutions for these limitations. For instance, Tong et al. [45] discovered that CLIP vision encoders, employed by most VLMs, often fail to distinguish between certain image pairs despite their apparent visual differences. They suggested utilizing alternative vision encoders, such as DINO, to address this problem. Our work aligns with these studies in understanding VLM limitations and proposing solutions. We find that current VLMs struggle with image classification, and we thoroughly investigate the reasons behind this, proposing a simple solution to integrate classification-focused data into VLM training.

**Image classification.** Image classification is one of the most fundamental capabilities of machine intelligence. Starting in the 1990s, researchers collected datasets like MNIST [25] for digit classification and CIFAR [22] for object classification, using various machine learning algorithms such as SVMs and MLPs to tackle the problem. A significant milestone was the ImageNet [11], which elevated the scale and quality of datasets to a new level, driving advancements in deep learning. With the rise of deep supervised and self-supervised learning [23, 9], performance on these datasets soon began to saturate. As a result, classification models have superseded most human labelers, leading researchers to focus on more advanced visual intelligence tasks like visual reasoning. In this work, we revisit this simple yet fundamental task, discovering that current VLMs struggle with it. We demonstrate that classification remains essential, as it forms the foundation for more advanced capabilities, and enhancing VLMs' classification capabilities improves their overall performance.

## 6 Conclusion

In this work, we explored the use of visually-grounded language models (VLMs) as image classifiers. We found that their performance is limited across various datasets. We then analyzed the reasons behind these limitations and, based on our findings, trained a VLM with enhanced general capabilities.

## Acknowledgements

We thank Irena Gao, Jeffrey Gu, Weixin Liang, Alejandro Lozano, Shiori Sagawa, Ruocheng Wang, Yunzhi Zhang, Orr Zohar for providing valuable feedback to this project. This work is in part supported by the NSF AI Institute for Foundations of Machine Learning (IFML), Open Philanthropy, and the Allen Institute for AI. Serena Yeung-Levy is a Chan Zuckerberg Biohub — San Francisco Investigator.

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

## Broader Impacts and Ethics Statement

In this work, we identify a core limitation of visually-grounded language models (VLMs) and propose a simple solution to address this issue. We find that VLMs underperform in image classification. By integrating classification data into VLM training, we have demonstrated that this intervention not only enhances classification accuracy but also improves the general capabilities of VLMs. This advancement holds promise for real-world applications, such as virtual assistants for visually impaired individuals. However, there are potential risks associated with incorporating race-based or gender-based classification data, which could amplify discriminatory biases in VLMs. We strongly emphasize the importance of ethical and responsible use of our approach to ensure it contributes positively to our digital ecosystem.

## Reproducibility Statement

We provide an open-source implementation of our work and have released the ImageWikiQA dataset at `https://github.com/yuhui-zh15/VLMClassifier`. This will enable researchers to reproduce all experiments described in this paper and conduct their own analyses on additional VLMs and datasets.

## Limitations

Due to computational constraints, we cannot evaluate all possible VLMs and datasets. We select two representative ones, LLaVA and BLIP, and conduct analyses on four datasets. Additionally, as we find that VLMs have information in their latent space, we conduct large-scale training to decode them. Developing a zero-shot method to decode this information without extensive training could be a promising direction for future research.

## Compute Resources

We use four NVIDIA L40S GPUs for all experiments. Most inference experiments take 5-20 hours on a single L40S GPU. The most computationally intensive step is fine-tuning. Fine-tuning `LLaVA1.5-7B` on 1.28M ImageNet data takes 1.4 days for one epoch using four GPUs. Fine-tuning `LLaVA1.5-7B` on 1.28M ImageNet data combined with 665K original LLaVA instruction-tuning data takes 3.5 days for one epoch using four GPUs. For the 13B model, the cost is approximately 40% higher.

## Summary of Appendix

We provide additional details for each section in the main paper.

- In §A, we provide details of models and data for §2.
- In §B, we provide details of experimental settings and results for §3.
- In §C, we provide details of our collected ImageWikiQA dataset for §4.
- In §D, we summarize the contributions of our work.

## A  Supplementary for Section 2

### A.1  Model

We include model details in Table 6.

### A.2  Data

We include data details in Table 7.

| Model | Link | Property | Version |
|---|---|---|---|
| GPT4 [36] | https://platform.openai.com/docs/guides/vision | Proprietary | gpt-4-turbo-2024-04-09 |
| GeminiPro [44] | https://ai.google.dev/ | Proprietary | gemini-1.0-pro-vision-001 |
| Claude3 [3] | https://console.anthropic.com/ | Proprietary | claude-3-opus-20240229 |
| LLaVA1.5-7B [32] | https://huggingface.co/liuhaotian/llava-v1.5-7b | Public | - |
| LLaVA1.5-13B [32] | https://huggingface.co/liuhaotian/llava-v1.5-13b | Public | - |
| LLaVANeXT-V7B [31] | https://huggingface.co/liuhaotian/llava-v1.6-vicuna-7b | Public | - |
| LLaVANeXT-M7B [31] | https://huggingface.co/liuhaotian/llava-v1.6-mistral-7b | Public | - |
| BLIP2-2.7B [27] | https://huggingface.co/Salesforce/blip2-opt-2.7b | Public | - |
| IBLIP-7B [10] | https://huggingface.co/Salesforce/instructblip-vicuna-7b | Public | - |
| IBLIP-13B [10] | https://huggingface.co/Salesforce/instructblip-vicuna-13b | Public | - |
| CLIP-L [37] | https://github.com/openai/CLIP | Public | - |
| EVA-G [43] | https://github.com/baaivision/EVA | Public | - |

Table 6: **Model details.**

| Dataset | Link | Training | Validation | # Classes |
|---|---|---|---|---|
| ImageNet [11] | https://www.image-net.org/ | 1.28M | 50K | 1000 |
| Flowers102 [35] | https://www.tensorflow.org/datasets/catalog/oxford_flowers102 | 2.0K | 6.1K | 102 |
| StanfordCars [21] | https://www.tensorflow.org/datasets/catalog/cars196 | 8.1K | 8.0K | 196 |
| Caltech101 [13] | https://www.tensorflow.org/datasets/catalog/caltech101 | 4.3K | 4.3K | 101 |

Table 7: **Data details.**

| Model | Accuracy w/o Synonyms | Accuracy w/ Synonyms | Improvement ($\Delta$) |
|---|---|---|---|
| BLIP2-2.7B | 25.3 | 27.8 | 2.5 |
| IBLIP-7B | 14.6 | 16.5 | 1.9 |
| IBLIP-13B | 14.7 | 16.6 | 1.9 |
| LLaVA1.5-7B | 22.8 | 24.6 | 1.8 |
| LLaVANeXT-V7B | 29.4 | 32.2 | 2.8 |
| LLaVA1.5-13B | 24.3 | 26.0 | 1.7 |
| LLaVANeXT-M7B | 32.3 | 35.1 | 2.8 |
| Claude3 | 53.6 | 56.3 | 2.7 |
| GeminiPro | 39.2 | 42.5 | 3.3 |
| GPT4 | 48.5 | 51.1 | 2.6 |

Table 8: **Model accuracy with and without label synonyms on ImageNet, and corresponding improvement.**

### A.3 Synonymous Text Labels in ImageNet Evaluation

In ImageNet, many classes have multiple valid textual labels. For instance, the class identifier n01496331 corresponds to both "electric ray" and "crampfish". To address this, we incorporated label synonyms[3] in our evaluation. We treat the classification as correct if any textual label is included in the VLM generation. Including synonyms improves accuracy by only 1%-3% (Table 8), still leaving a notable performance gap when compared to CLIP.

# B    Supplementary for Section 3

## B.1    Prompt Variation Detail

**Prompt details.**    We use three different prompts with varying wordings. Base Prompt: "What type of object is in this photo?" Prompt Alternative 1: "Identify the object in this image." Prompt Alternative 2: "What is the main object depicted in this photograph?"

**Label order details.**    For the fixed order, we use the default dataset label order for each image.

---

[3]https://gist.github.com/yrevar/942d3a0ac09ec9e5eb3a

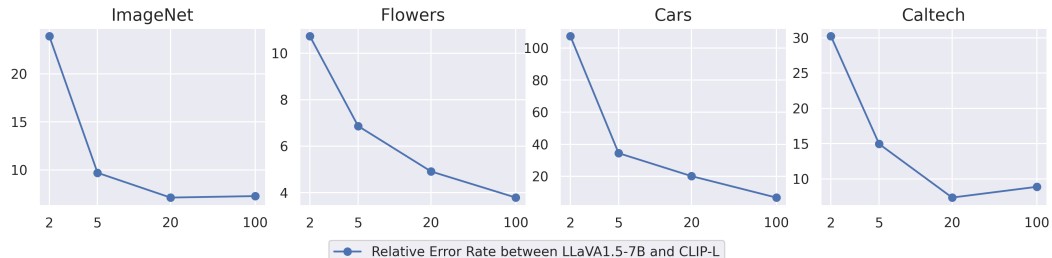

Figure 4: **Analysis of the label set size.** For each image, we randomly sample 100, 20, 5, and 2 candidate classes from the entire set of classes. While the absolute performance gap between VLMs and CLIPs decreases as the number of classes is reduced, the relative performance gap increases. The X-axis represents the number of classes, and the Y-axis represents the relative error rate between `LLaVA1.5-7B` and `CLIP-L`.

| Model | # Classes | 🎯 | 🌻 | 🚗 | Ⓒ |
|---|---|---|---|---|---|
| LLaVA1.5-7B | 2 | 94.3 | 67.1 | 78.5 | 96.4 |
|  | 5 | 92.3 | 50.5 | 67.9 | 96.6 |
|  | 20 | 82.0 | 29.6 | 22.0 | 91.8 |
|  | 100 | 46.6 | 10.0 | 1.0 | 62.1 |
| BLIP2-2.7B | 2 | 14.3 | 17.4 | 0.9 | 29.6 |
|  | 5 | 11.9 | 16.9 | 1.3 | 26.7 |
|  | 20 | 8.0 | 12.2 | 1.5 | 20.9 |
|  | 100 | 5.5 | 11.2 | 3.9 | 22.4 |
| CLIP-L | 2 | 99.8 | 96.9 | 99.8 | 99.9 |
|  | 5 | 99.2 | 92.8 | 99.1 | 99.8 |
|  | 20 | 97.5 | 85.7 | 96.1 | 98.9 |
|  | 100 | 92.7 | 76.3 | 85.4 | 95.7 |

Table 9: **Analysis of the label set size.** This is the table version of Figure 2.

## B.2 Label Set Size Details

**Performance details.**    The table version of Figure 2 is provided in Table 9. By reducing the number of labels for classification, we can narrow the gap between VLM and CLIP, but a gap remains across all label set sizes, even with just two labels (two-way classification).

**Relative performance gaps.**    We find that while the absolute gap between VLMs and CLIPs narrows with reduced label size, the relative gap increases. For example, in two-way classification on ImageNet, VLMs have a 5.7% error rate, while CLIP has a 0.2% error rate, resulting in a 28.5x gap; for 20 classes, VLMs have an 18.0% error rate, while CLIP has a 2.5% error rate, resulting in a 7.2x gap. This trend is indicated by the error rates between VLMs and CLIP shown in Figure 4.

**Larger gap on fine-grained classification datasets.**    The absolute gap on Flowers and Cars is larger than ImageNet and Caltech when reducing the label set size. This may be because Flowers and Cars are more fine-grained classification datasets, while ImageNet and Caltech are more coarse-grained. VLMs are weaker in fine-grained classification compared to CLIP, resulting in a larger gap.

## B.3 Inference Strategy Details

**Experimental details.**    Probabilistic inference methods require separate computations for each class, which introduces significant computational costs. For example, on ImageNet with 1000 classes, each example takes 1000 times longer to process than with a direct generation pipeline. Therefore, we randomly sampled 1000 examples from each dataset's validation set for evaluation.

| Position | 0 | 100 | 200 | 300 | 400 | 500 | 600 | -3 | -2 | -1 (Last) | Average |
|----------|-----|------|------|------|------|------|------|------|------|-----------|---------|
| Accuracy | 0.7 | 17.1 | 18.1 | 25.0 | 25.8 | 23.7 | 52.9 | 84.0 | 85.8 | 94.5 | **96.2** |

Table 10: **Probing the last token or the average token results in much better performance than probing other token positions.** Experiments are done using `LLaVA1.5-7B` on the Flowers102 dataset.

## B.4 Information Loss Details

**CLIP global vs. local features.** The global feature of CLIP is commonly used in traditional classification settings, which has 1 token for each image. However, VLMs utilize local patch features for classification, which has 576 tokens for each image using `CLIP-L`. One might hypothesize that local features contain less information than global features. In practice, VLMs use the second-to-last layer of CLIP local features rather than the last layer. Since the last layer's global feature is the weighted average of the second-to-last layer's features through self-attention computation, we can theoretically conclude that local features have at least the same level of information as CLIP global features.

**Feature extraction details.** We extract features from VLM's last layer by feeding the LLaVA default prompt "USER: <576 Image Tokens> What type of object is in this photo? ASSISTANT:" into the LLaVA, or BLIP default prompt "<Image Tokens> Question: What type of object is in this photo? Answer:" into the BLIP. We use either the last token feature (i.e., the token ":") or the average feature over all the tokens.

**Linear probing details.** We train the linear layer on the training set with a batch size of 512, a learning rate of 1e-3 using the Adam optimizer, and for 500 epochs. After training, we report the best performance on the validation set.

**Probing position.** In practice, we find that probing the last token or the average token results in much better performance than probing other token positions, as shown in Table 10. We leave this as an interesting question for future research.

## B.5 Training Objective Detail

**LLaVA fine-tuning details.** We convert each image and class label into the text format using the LLaVA default template "USER: <576 Image Tokens> What type of object is in this photo? ASSISTANT: <Class Name>." We conduct two settings for fine-tuning LLaVA. In the first setting, we only fine-tune the MLP projector between CLIP and the language model (LM). The projector is trained on the training set with a batch size of 64 and a learning rate of 2e-5 using the AdamW optimizer for 50 epochs (1 epoch for ImageNet), with a warmup ratio of 0.03. In the second setting, we fine-tune both the MLP projector and the LM using LoRA. The projector and LM are trained on the training set with a batch size of 64, a learning rate of 2e-5 for the projector and 2e-4 for the LM, using the AdamW optimizer for 50 epochs (1 epoch for ImageNet), with a warmup ratio of 0.03, a LoRA rank of 128, and a LoRA alpha of 256. For both settings, we report the best performance on the validation set after training. Fine-tuning on ImageNet for 1 epoch requires 130 L40 GPU hours.

**BLIP fine-tuning details.** We convert each image and class label into the text format using the BLIP default template "<Image Tokens> Question: What type of object is in this photo? Answer: <Class Name>." We train the Q-former projector between CLIP and the LM on the training set with a batch size of 64, a learning rate of 2e-5, and a weight decay of 0.05 using the AdamW optimizer for 100 epochs (10 epochs for ImageNet), with 1000 warmup steps. After training, we report the best performance on the validation set. Fine-tuning on ImageNet for 10 epochs requires 120 L40 GPU hours.

**CLIP fine-tuning details.** We initialize a linear layer on top of the CLIP image encoder. The CLIP model is frozen, and only the linear layer is trained. The linear layer is trained on the training set with a batch size of 512, a learning rate of 1e-3, and no weight decay using the Adam optimizer for 300 epochs (40 epochs for ImageNet). After training, we report the best performance on the validation set.

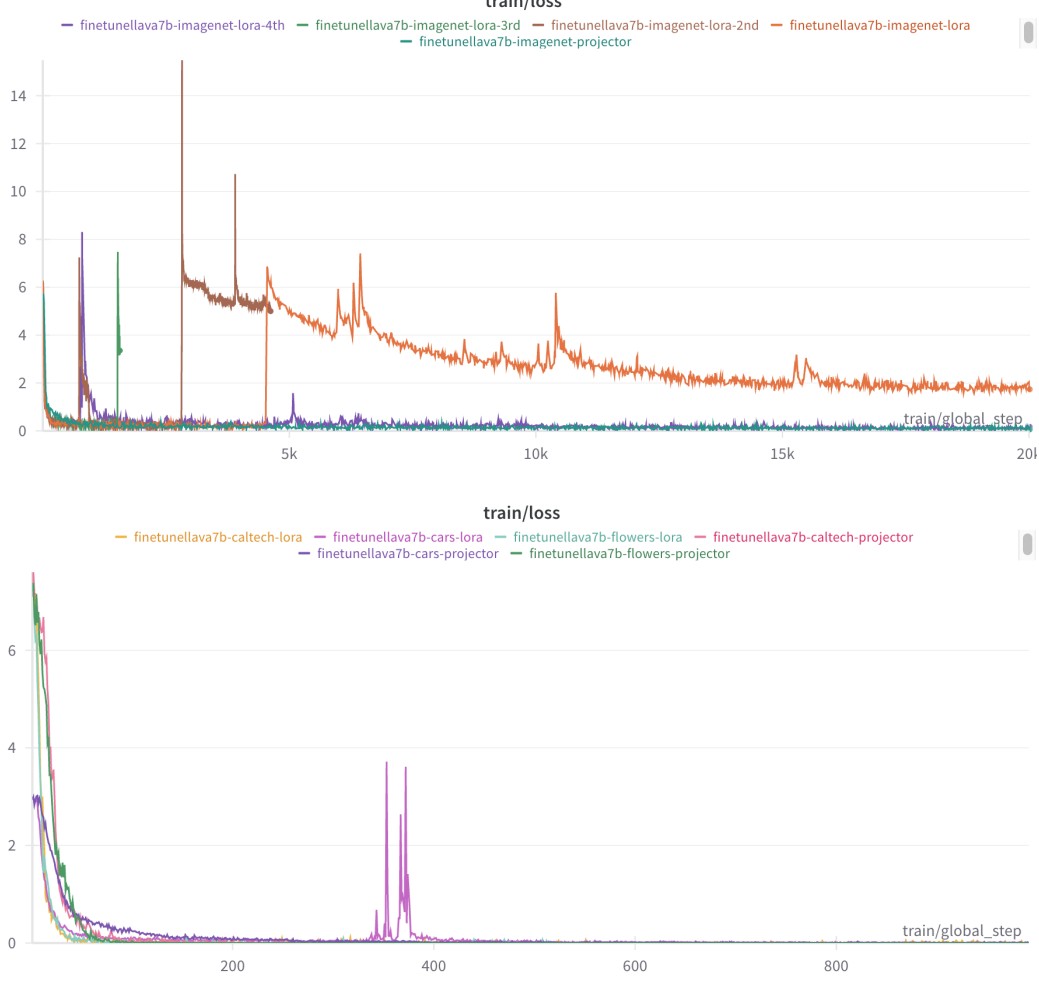

Figure 5: **Fine-tuning only the projector improves numerical stability.** *(Top)* Fine-tuning LLMs with LoRA often results in numerical instabilities, manifesting as spikes in loss (purple, green, brown, orange curves). In contrast, fine-tuning only the projector leads to a consistently steady decrease in loss (teal curve). Despite experimenting with various hyperparameters for ImageNet, the instability remained. *(Bottom)* Occasionally, the spikes normalize with continued training. Here, we present an example using the StanfordCars dataset (pink curve).

| | Spearman Correlation | | | | Pearson Correlation | | | |
|---|---|---|---|---|---|---|---|---|
| | 🧩 | 🌻 | 🚗 | ⓒ | 🧩 | 🌻 | 🚗 | ⓒ |
| CLIP-L | -0.13 | 0.30 | 0.04 | -0.03 | -0.08 | 0.08 | 0.06 | 0.00 |
| LLaVA1.5-7B Fine-tuned on ImageNet | -0.02 | -0.03 | 0.04 | 0.30 | -0.05 | 0.08 | 0.04 | 0.18 |
| LLaVA1.5-7B | **0.76** | **0.40** | N/A | **0.69** | **0.35** | **0.64** | N/A | **0.27** |

Table 11: **Correlation between class frequency and model's accuracy on the class.** This supplements Figure 3.

**Numerical instability.**    When fine-tuning both the projector and the LM with LoRA for LLaVA, we experience significant numerical instability. On ImageNet, despite adjusting three different hyperparameters such as the learning rate and batch size, the loss consistently peaks during training and does not recover. On the StanfordCars dataset, numerical instability is also observed, but the loss returns to normal during training. This can be seen in Figure 5.

| Method | LLaVA1.5-7B | | | | GPT4 | | | |
|---|---|---|---|---|---|---|---|---|
| | 🗂️ | 🌻 | 🚙 | Ⓒ | 🗂️ | 🌻 | 🚙 | Ⓒ |
| **Prompt Variation** | | | | | | | | |
| Base Prompt | **22.8** | 5.9 | 0.0 | 47.1 | 48.5 | 51.0 | 0.1 | 61.0 |
| w/ Label (Fixed Order) | N/A | 6.1 | 0.1 | **70.5** | 61.2 | 79.0 | 57.1 | 91.2 |
| w/ Label (Random Order) | N/A | 10.2 | 0.0 | 62.1 | 60.6 | **79.9** | 58.2 | **94.2** |
| w/ Label (Random Order) + CoT | N/A | **18.1** | **0.4** | 64.5 | **62.2** | 79.0 | **59.8** | 94.0 |

Table 12: **Analysis of VLMs from the inference perspective.** Compared to Table 2, we added the results for the proprietary `GPT4` model. We explore prompt variation such as label order and chain-of-thought (CoT) and find it has a limited impact on the performance, which applies to proprietary VLMs such as GPT4.

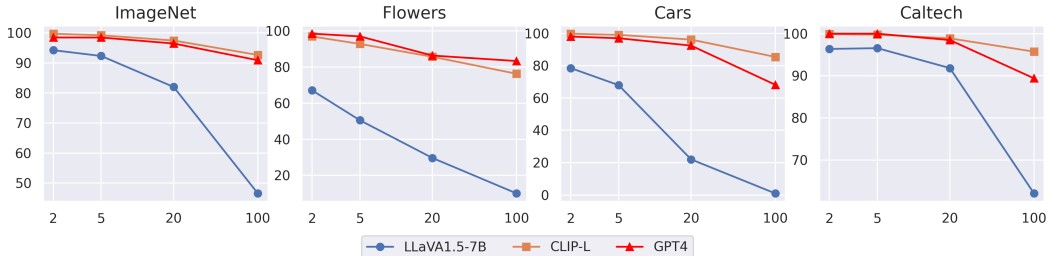

Figure 6: **Analysis of the label set size.** Compared to Figure 2, we added the line for the proprietary `GPT4` model. `LLaVA1.5-7B` uses `CLIP-L` as its vision encoder, while `GPT4` presumably employs a superior vision encoder. Therefore, comparing `GPT4` to `CLIP-L` is not entirely fair. However, we observe that `GPT4` performs worse than `CLIP-L` on ImageNet even when the number of classes is reduced to 2.

## B.6 Data Detail

**Data processing details.** The training of LLaVA comprises two stages: pre-training and instruction tuning. The pre-training stage utilizes noisy image captioning data, whereas the instruction tuning stage employs meticulously curated multi-turn conversation data. We tokenize all sentences within the data, compute the frequency for each class, and then calculate the accuracy for each class. This results in 1000 (frequency, accuracy) pairs for ImageNet's 1000 classes. This process is similarly applied to all other datasets.

**Correlation results.** We compute the Spearman and Pearson correlations between frequency and accuracy of zero-shot `LLaVA1.5-7B`. The results are presented in Table 11. For comparison, we conducted the same analysis for `CLIP-L` and fine-tuned `LLaVA1.5-7B`.

**Correlation for two stages.** Given that LLaVA training involves two stages, we examine whether pre-training or instruction-tuning data has a greater impact on classification performance. We compute the Spearman and Pearson correlations separately for pre-training and instruction-tuning data. Our findings indicate that the correlation is similar for both stages. For instance, on ImageNet, the Spearman and Pearson correlations are 0.73 and 0.34 for pre-training, and 0.76 and 0.33 for instruction tuning. These values are close to the combined data correlations, which are 0.76 and 0.35 for Spearman and Pearson correlations, respectively, as reported in Table 11.

## B.7 Analysis of Proprietary VLMs

In the main paper, we analyzed two public VLMs about why they are bad at image classification. For proprietary VLMs, they haven't released the training details and data usage, so it is impossible to directly address training-related or data-related hypotheses.

For inference-related hypotheses, we conduct additional experiments with GPT4 (Table 12, Figure 6), and the conclusion is the same as the public VLMs. For example, GPT-4 achieves 60.6% accuracy on ImageNet without CoT. With CoT, GPT-4 achieves 62.2% accuracy, demonstrating that CoT has a limited impact on image classification, even for large VLMs.

For training-related and data-related hypotheses, recent work PaliGemma [6] from DeepMind shows that most VLM tasks benefit significantly from longer pre-training with more data (in its Figure 4 and Appendix K). This aligns with our main conclusion that data is the critical factor in improving VLM performance.

## B.8 Open-world vs. Closed-world Setting

§3 and §4 all use "open-world" settings (not providing the label set in the prompt), except for prompt variation analysis and label set size analysis in §3.1, using "closed-world" settings (providing the label set in the prompt).

Naturally, the "closed-world" setting narrows the model's generation space and increases accuracy. When the model can easily predict the class in the label set by modifying the inference space (probabilistic inference in §3.1 and probing VLM in §3.2) or training VLMs with more classification data (§4), the advantage of the "closed-world" setting doesn't exist. Thus, for these experiments, we use the "open-world" setting.

## B.9 Number Comparison of Table 1 vs. Table 2

Table 2's "Base Prompt w/ Label (Random Order)" is equivalent to Table 1's "Closed-World Setting". In this setting, we concatenate label candidates in a random order for each question: "What's in the image? Choose one from random_shuffle([cat, dog, pig])". The accuracy for these two settings is identical between Table 2 and Table 1.

Table 2's "Base Prompt w/ Label (Fixed Order)" concatenates label candidates in a fixed order for each question: "What's in the image? Choose one from [cat, dog, pig]". This setting is used to rule out the possibility that the order of labels might affect model performance.

## B.10 Why Do VLMs Have Information but Cannot Decode?

In 3, we find that VLMs have the essential information for classification. If the VLMs have the information, then why are they not able to decode it?

One possible reason is that the VLM's decoding space is too large and not aligned with the visual features. Specifically, VLM decoding is performed through next-word prediction, which usually involves a vocabulary of over 10,000 words. The output text embedding is not aligned with the visual features due to insufficient data to align these spaces.

In general, having information in a model does not necessarily mean the model can express that information. This phenomenon is also observed in other research areas. For example, after training ResNets or ViTs with self-supervised learning methods like SimCLR, the model acquires discriminative information for different classes. However, this information can only be decoded by adding and training a linear layer on a new dataset.

Similarly, we demonstrate that VLMs possess classification information, but it is not readily expressible. Adding classification-related data helps bridge the gap between "information possession" and "information expression" (refer to Table 3).

## B.11 Why Does Fine-tuning Only on ImageNet Get Worse Performance on ImageWikiQA?

In Table 5, the "Finetuned on ImageNet" model performance (30.6) is worse than the non-finetuned model performance (38.0). This is because fine-tuning often improves in-distribution performance but does not necessarily enhance out-of-distribution or general capabilities.

Table 5 presents the results of models fine-tuned on ImageNet and evaluated on ImageWikiQA, which are vastly different datasets. ImageNet questions only require classification, such as "Q: <image> What is in the image? A: dog," while ImageWikiQA questions demand both classification and knowledge/reasoning, such as "Q: <image> What is the native region of this object? A: South America."

Fine-tuning solely on ImageNet trains the model to classify, but it can lead to a loss of general capabilities like reasoning and knowledge, resulting in lower performance on ImageWikiQA (30.6 fine-tuned vs. 38.0 pre-trained). This phenomenon is known as "catastrophic forgetting" [19].

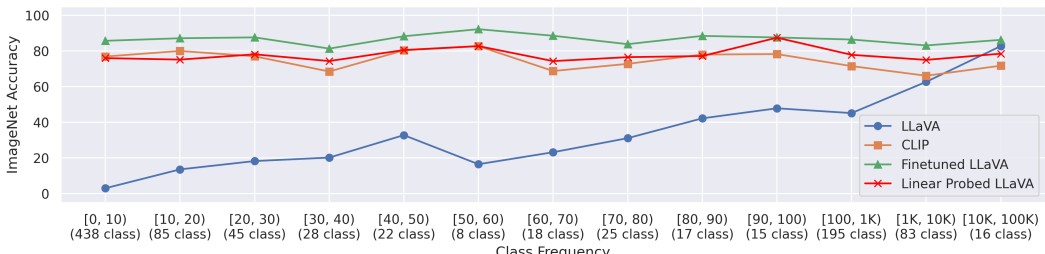

Figure 7: **Analysis of linear probed VLM from the data perspective.** Compared to Figure 3, we added the line for the linear probed LLaVA model, which has zero correlation. The zero correlation is expected because probing on the ImageNet dataset equalizes class frequencies, as ImageNet classes are evenly distributed.

However, when we fine-tune on both ImageNet and instruction tuning data, the model learns classification while retaining its original capabilities, leading to a significant performance improvement on ImageWikiQA (49.8 fine-tuned vs. 38.0 pre-trained).

## B.12    Analysis of Linear Probed VLM from Data Perspective

Figure 7 plots results for the linear probed LLaVA model, which aligns closely with the fine-tuned LLaVA. This alignment shows a zero correlation with class frequency since fine-tuning or probing on the balanced ImageNet dataset equalizes class frequencies, where each class is evenly represented.

# C    Supplementary for Section 4

## C.1    ImageWikiQA Details

**Data collection details.**    ImageWikiQA only contains a test set, providing a valuable benchmark for assessing both fine-grained classification and knowledge-based reasoning in VLMs.

The ImageWikiQA dataset was constructed in two stages. In the first stage, we generated questions at the class level for each of the 1,000 ImageNet classes. For instance, questions like "What is the native region of <class name>?" were created to capture high-level class information. In the second stage, we refined these questions at the image level. For each ImageNet class, we randomly sampled 3 images from the 50 available images from the ImageNet validation set and formulated specific questions like "What is the native region of <image i>?". Each question in ImageWikiQA is paired with a single image, ensuring only one image per question.

The ImageWikiQA dataset contains 2,000 unique questions. A question like "What is the native region of <image i>?" is counted multiple times if it is asked for different images or classes. For example, if the question is asked for 2 cat images and 2 dog images, it would count as 4 questions in total. To avoid over-representing any particular question, a question like "What is the native region of <image i>?" is limited to a maximum of 3 occurrences across the 2,000 questions.

**Data generation prompt.**    We provide the prompt used to generate ImageWikiQA in Figure 8 and the prompt used to filter ImageWikiQA in Figure 9.

**More examples.**    We provide more examples of the ImageWikiQA dataset in Table 14.

## C.2    Results Details

**LLaVA fine-tuning details.**    For ImageNet, we convert each image and class label into the text format using the LLaVA default template "USER: <576 Image Tokens> What type of object is in this photo? ASSISTANT: <Class Name>." Then, we concatenate the original LLaVA 665K instruction tuning dataset with the 1.28M ImageNet classification dataset. Due to numerical stability concerns, we train only the projector layer of LLaVA. The projector is trained on the combined dataset with a batch size of 64, a learning rate of 2e-5, using the AdamW optimizer, for 1 epoch, with a warmup ratio of 0.03. The training process takes approximately 340 hours on a single L40 GPU.

```
Come up with five multiple-choice questions for a {classname} to
    ↪ examine the knowledge of an expert.

The questions should come from the following Wikipedia articles on
    ↪ {classname}:
```
{wikipedia page}
```

**Instruction**:
Each question should have four choices, one of which is the correct
    ↪ answer.
Note that the Wikipedia articles will not be accessible to the
    ↪ test-takers, so please do not reference specific details from
    ↪ the articles in the questions.
Use "this object" rather than "{classname}" in the questions. We want
    ↪ to give test-takers an image of this object, not the word
    ↪ itself.
Each question should have four fields: "question" (str), "choices"
    ↪ (list[str]), "answer" (int, starting from 0)), and "reference"
    ↪ (str, original sentences from Wikipedia articles).
Output in JSON format.
```

Figure 8: **Prompt used to generate ImageWikiQA using GPT4.**

```
Answer the multiple-choice question below.
{question with ground-truth class name}
A. {choice A}
B. {choice B}
C. {choice C}
D. {choice D}
Output only one character A/B/C/D.

-----

Answer the multiple-choice question below.
{question without ground-truth class name}
A. {choice A}
B. {choice B}
C. {choice C}
D. {choice D}
The question mentions "this object". However, we don't know what the
    ↪ object is. Try your best to guess the answer without knowing
    ↪ the object.
Output only one character A/B/C/D.
```

Figure 9: **Prompt used to filter ImageWikiQA using GPT4.**

## C.3 Performance of Fine-tuned VLM on Other Datasets

To understand whether incorporating the ImageNet 1.28M classification data into the original LLaVA instruction-tuning data will harm the model's general capability, we further evaluated the performance of our fine-tuned visually-grounded language model (VLM) on additional benchmarks to ensure its robustness across a variety of tasks. In particular, we tested the model on TextVQA [42], POPE [29], and MMVet [51]. The results, presented in Table 13, show that the fine-tuned LLaVA1.5-7B maintains almost identical performance compared to the original version.

This result aligns with expectations since our fine-tuning process incorporated all of the original LLaVA training data during instruction tuning. For instance, on TextVQA, the fine-tuned model achieved an accuracy of 58.0%, closely matching the original model's 58.2%. Additionally, the fine-tuned model slightly outperformed the original on POPE Popular and POPE Adverse benchmarks, showing small improvements of 0.2% and 0.3%, respectively. Performance on MMVet remained

| Model | TextVQA | POPE Popular | POPE Adverse | MMVet |
|---|---|---|---|---|
| LLaVA1.5-7B (Official Released) | 58.2 | 86.1 | 84.2 | 31.1 |
| LLaVA1.5-7B (Further Finetuned) | 58.0 | 86.3 | 84.5 | 31.1 |

Table 13: **Performance of LLaVA1.5-7B before and after fine-tuning on TextVQA, POPE, and MMVet datasets.** Fine-tuning resulted in consistent performance across all benchmarks.

unchanged at 31.1%. These findings demonstrate that the fine-tuning process preserves the model's strengths across these varied benchmarks.

## C.4 Low Accuracy on ImageWikiQA with Ground-truth Class Names

Table 5 presents the accuracy on ImageWikiQA, which includes questions derived from Wikipedia requiring extensive world knowledge (e.g., "What is the native region of the guinea pig?").

The modest accuracy of 55.9% by LLaVA1.5-7B with provided ground-truth class names reflects its limited world knowledge for ImageNet classes. This limitation aligns with findings that smaller models, like Vicuna-7B, lack world knowledge compared to larger models like GPT-4.

## C.5 Other Fine-Tuning and Inference Strategies

Other fine-tuning or inference modifications, such as applying a linear probing loss in the VLM output space during training or using k-nearest neighbors (KNN) in the VLM output space for evaluation, can also enhance VLM classification performance.

However, our primary goal is to improve VLMs' general capabilities across a range of tasks, not just classification. The universal inference interface for various tasks is text generation, whereas approaches like KNN-based inference are tailored solely to classification and do not generalize to other task types. For fine-tuning, we experimented with adding a new token and applying a linear probing loss. This approach did not yield improvements on ImageWikiQA, indicating that task-specific fine-tuning does not enhance the overall capabilities of VLMs.

In summary, without changing VLM training or inference to maintain its general capabilities for different tasks, adding data is the most promising and effective approach.

# D Summary of Contributions

Object recognition is fundamental to the general capability of visually-grounded language models (VLMs), yet current VLMs perform poorly in this area. We are the first to thoroughly investigate this critical issue and propose a potential solution to improve it.

**Our primary contribution lies in identifying the problem (i.e., VLMs are inadequate image classifiers, a significant weakness that has been overlooked) and conducting an in-depth investigation (i.e., understanding why VLMs are poor image classifiers and how to address this issue).**

In summary, our contributions are threefold:

- **Thorough Evaluation to Identify the Problem:** We thoroughly evaluated current public and proprietary VLMs on four common classification benchmarks and discovered that their performance significantly lags behind CLIP. This finding is counterintuitive because VLMs often use CLIP as their vision encoders. **This finding reveals a weakness in VLMs that previous works have not widely noted [2, 49]. Understanding the limitations of VLMs is crucial given their increasing deployment in various scenarios.**

- **Hypothesis Testing to Understand the Problem:** Given the poor performance of VLMs in classification, we investigated the underlying reasons. We considered multiple plausible hypotheses related to VLM inference, training, and data perspectives. For example, **essential information for classification could be lost during the vision encoder's propagation through multiple LLM layers; VLMs might be inherently poor at classification due to their text generation training objective compared to the standard cross-entropy objective.** Our thorough investigation ruled

out these reasons but revealed the major issue: the lack of alignment data. **This finding is counterintuitive because other alternative hypotheses also seemed plausible.**

- **Improving VLMs based on Our Understanding:** Based on our findings, we explored ways to enhance VLM performance. We believe that classification is foundational to more advanced capabilities. For example, if a VLM cannot accurately classify mushroom species, it will also struggle with follow-up questions, such as whether a particular mushroom is poisonous. Indeed, **we found that simply adding classification data not only improves VLMs' classification performance but also their general capabilities, demonstrating that accurately identifying objects is a prerequisite for answering complex questions about these objects.**

**Impact:**   Recent work from Google DeepMind, PaliGemma [6], supports our main conclusion that data is the critical factor in improving VLM performance. They also found that most VLM tasks benefit significantly from longer pre-training with more data (Appendix K). **This demonstrates that our analysis can inspire researchers to build better VLMs in the future.**

| Image | Question | Choices | Reference Wikipedia |
|---|---|---|---|
| | Which position is particularly important for this object when transporting a patient in shock? | A. Fowler's position
B. Upright position
C. Trendelenburg position
D. Flat position | The feet can be raised to what is called the Trendelenburg position, indicated for patients in shock. |
| | Which characteristic is NOT typical for the temperament of this object? | A. Very intelligent and loyal
B. Easily bored and highly energetic
C. Quiet and suspicious of strangers
D. Highly aggressive and loud | Giant Schnauzers are usually a quiet breed... It has the potential to be aggressive, but Giant Schnauzers are usually reserved |
| | What is the traditional color combination of this object's coat? | A. Black with white markings
B. Solid red
C. Red with white markings
D. White with black markings | The breed's coat only comes in a single colour combination of white with red markings, usually in a piebald pattern. |
| | What steering mechanism is traditionally associated with this object when it possesses four wheels? | A. A rudder
B. Turntable or fifth wheel
C. Flap control
D. A steering wheel | A four-wheeled vehicle is also steered by the shafts or pole, which are attached to the front axle; this swivels on a 'turntable' or 'fifth wheel' beneath the vehicle. |
| | What is a notable behavior of this object during mating? | A. Males compete in physical combat
B. Releases a sound to attract mates
C. The female releases a pheromone to reduce aggression in males
D. Both sexes change color to a brighter hue | The female releases a pheromone which causes the males to become less aggressive and to begin courtship. |
| | Which of the following is known as the world's largest this object? | A. The LEGO Store
B. Hamleys
C. Toys "R" Us
D. FAO Schwarz | Notable Examples. - Hamleys, the world's largest toy shop |
| | In which habitats is this object predominantly found? | A. Low-altitude woodlands and forest edges
B. Polar and subpolar zones
C. Temperate forests and grasslands
D. Desert and arid regions | They are principally birds of low-altitude woodlands and forests, and particularly of forest edge and canopy. |
| | In which type of climates does this object predominantly thrive? | A. Mediterranean
B. Subarctic
C. Tropical rainforest
D. Desert | The plant tolerates seasonal drought, and the Middle Eastern and Mediterranean climates are especially suitable to it. |
| | Which of the following features was introduced by Philips in the development of this object? | A. Eight-to-fourteen modulation (EFM)
B. Diagonal error correction
C. Anti-shock buffering
D. Cross-interleaved Reed-Solomon Coding (CIRC) | Philips also contributed eight-to-fourteen modulation (EFM), which offers a certain resilience to defects such as scratches and fingerprints. |
| | What is a common contemporary use for this object in urban settings? | A. Display in museums
B. Storing chilled goods for retail
C. Transporting large amounts of heavy equipment
D. Delivering business mail across the city | Messenger bag: one long strap worn across the body, inspired by bags worn by urban messengers to deliver business mail, a modern silhouette |

Table 14: **Examples of ImageWikiQA.** Each example has an image, question, four choices (correct choice highlighted in orange), and reference Wikipedia sentences.

