# OpenReview forum: "Why are Visually-Grounded Language Models Bad at Image Classification?"
_NeurIPS.cc/2024/Conference — NeurIPS 2024 poster_

### Official Review · Reviewer_CuMn · 2024-06-30

**Soundness:** 3
**Presentation:** 3
**Contribution:** 3
**Rating:** 3
**Confidence:** 4

**Summary:**

This article notes that current top-performing VLMs, like GPT-4V and LLaVA, are unable to perform image classification tasks like the CLIP model. Despite having a larger number of parameters and incorporating a vision encoder from a pre-trained CLIP model.

This author thinks the main reason for this underperformance is data-related. Critical information for image classification is encoded in the VLM's latent space but can only be effectively decoded with enough training data.

The author proposes to incorporate classification-specific datasets (such as ImageNet) into VLM training; its performance in classification and complex visual tasks can be significantly improved. For example, after fine-tuning ImageNet, VLM's performance on the ImageWikiQA dataset improved by 11.8%.

**Strengths:**

The article clearly points out the performance issues of the current Visual Language Model (VLM) in image classification tasks. It systematically analyzes the possible reasons, filling the research gap in this field. This author thinks the main reason for this underperformance is data-related. By incorporating classification-specific datasets (such as ImageNet) into VLM training, the performance of classification and complex visual tasks can be significantly improved. For example, after fine-tuning ImageNet, VLM's performance on the ImageWikiQA dataset improved by 11.8%.

**Weaknesses:**

The novelty is limited.
This article assesses how well current VLMs perform on classification datasets and fine-tunes them using the same datasets.
These contributions are insufficient to warrant publication of this paper.

**Questions:**

I would like to know how well the fine-tuned VLMs perform on the original benchmarks such as MME, TextVQA, and others.
Can we achieve similar results if a classification dataset is included in the second stage of training with LLaVA?

**Limitations:**

Yes, the author has addressed relevant limitations and discussed the broader impacts adequately.

---

> ### Author Rebuttal · Authors · 2024-08-05
>
> We thank Reviewer CuMn for providing thoughtful feedback on our work. We address Reviewer CuMn’s questions below.
>
> ---
>
> **Limited novelty**
>
> > The novelty is limited. This article assesses how well current VLMs perform on classification datasets and fine-tunes them using the same datasets. These contributions are insufficient to warrant publication of this paper.
> >
>
> We want to clarify that our paper is not a method paper proposing new techniques to achieve better performance on standard benchmark leaderboards. Instead, it is an analysis paper.
>
> Object recognition is fundamental to the general reasoning capability of VLMs, yet current VLMs perform poorly in this area. We are the first to thoroughly investigate this critical issue and propose a potential solution to improve it.
>
> **Our primary contribution lies in identifying the problem (i.e., VLMs are inadequate image classifiers, a significant weakness that has been overlooked) and conducting an in-depth investigation (i.e., understanding why VLMs are poor image classifiers and how to address this issue).** This contribution goes far beyond merely assessing current VLMs on classification datasets and fine-tuning them using the same datasets.
>
> In summary, our contributions are threefold:
>
> 1. **Thorough Evaluation to Identify the Problem:** We thoroughly evaluated current public and proprietary VLMs on four common classification benchmarks and discovered that their performance significantly lags behind CLIP. This finding is counterintuitive because VLMs often use CLIP as their vision encoders. **This finding reveals a weakness in VLMs that previous works have not noted. Understanding the limitations of VLMs is crucial given their increasing deployment in various scenarios.**
> 2. **Hypothesis Testing to Understand the Problem:** Given the poor performance of VLMs in classification, we investigated the underlying reasons. We considered multiple plausible hypotheses related to VLM inference, training, and data perspectives. For example, **essential information for classification could be lost during the vision encoder’s propagation through multiple LLM layers; VLMs might be inherently poor at classification due to their text generation training objective compared to the standard cross-entropy objective.** Our thorough investigation ruled out these reasons but revealed the major issue: the lack of alignment data. **This finding is counterintuitive because other alternative hypotheses also seemed plausible.**
> 3. **Improving VLMs based on Our Understanding:** Based on our findings, we explored ways to enhance VLM performance. We believe that classification is foundational to more advanced capabilities. For example, if a VLM cannot accurately classify mushroom species, it will also struggle with follow-up questions, such as whether a particular mushroom is poisonous. Indeed, **we found that simply adding classification data not only improves VLMs’ classification performance but also their general capabilities, demonstrating that accurately identifying objects is a prerequisite for answering complex questions about these objects.**
>
> **Impact:** recent work from Google DeepMind, PaliGemma [1], supports our main conclusion that data is the critical factor in improving VLM performance. They also found that most VLM tasks benefit significantly from longer pre-training with more data (Figure 4, Appendix K). **This demonstrates that our analysis can inspire researchers to build better VLMs in the future.**
>
> [1] PaliGemma: A versatile 3B VLM for transfer.
>
> ---
>
> **Fine-tuned VLM performance on other benchmarks**
>
> > I would like to know how well the fine-tuned VLMs perform on the original benchmarks such as MME, TextVQA, and others. Can we achieve similar results if a classification dataset is included in the second stage of training with LLaVA?
> >
>
> Thank you for your question. **Following your suggestion, we evaluated both the original VLM and our fine-tuned VLM on three additional benchmarks and found that the VLM’s performance remain the same.**
>
> The table below shows the performance of the original LLaVA1.5-7B and the fine-tuned LLaVA1.5-7B on TextVQA, POPE, and MMVet. After fine-tuning, the LLaVA1.5-7B achieves the same accuracy as without fine-tuning. This result is intuitive because we included all the original LLaVA training data during our instruction tuning.
>
> We did not include MME because the dataset is not publicly available, and our access request has not yet been approved.
>
> We will release all the codes and model checkpoints to reproduce these results and include them in the revised paper.
>
> |  | TextVQA | POPE Popular | POPE Adverse | MMVet |
> | --- | --- | --- | --- | --- |
> | LLaVA1.5-7B (Official Released) | 58.2 | 86.1 | 84.2 | 31.1 |
> | LLaVA1.5-7B (Further Finetuned) | 58.0 | 86.3 | 84.5 | 31.1 |
>
> ---
>
> Thank you again for your feedback. Please let us know if you have further questions or concerns!

---

> ### Author Response · Authors · 2024-08-09
> **Gentle request for discussion**
>
> Dear Reviewer,
>
> We kindly request your feedback on our rebuttal. We believe we have thoroughly addressed your concerns regarding other VLM benchmarks and the novelty of our work.
>
> If our rebuttal has resolved your concerns, we would greatly appreciate it if you could reconsider your scores for our paper. Should you have any further questions or concerns, we are eager to discuss them.
>
> Thank you for your time and consideration!

---

> > ### Comment · Reviewer_CuMn · 2024-08-13
> >
> > Thank the authors for the response.
> >
> > After reading the rebuttal, I would like to keep the original rating.

---

> > > ### Author Response · Authors · 2024-08-13
> > >
> > > Thank you for your response! Could you clarify which of your concerns remain unaddressed? If you have any other questions or concerns, please feel free to let us know.

---

### Official Review · Reviewer_8fB5 · 2024-07-03

**Soundness:** 2
**Presentation:** 3
**Contribution:** 2
**Rating:** 5
**Confidence:** 4

**Summary:**

This paper analyzes the issue of large vision language models (VLMs) that perform poorly in common image classification datasets such as ImageNet. The authors analyze different perspectives on the problem, including trying different inference and training methods. For inference, the authors tried using different prompt variations, shrinking the number of classes, and computing conditional probabilities of class names. All these methods still leads to performance gaps between VLMs and CLIP. Furthermore, the authors confirm that sufficient class information is encoded in the output features of the visual encoder in VLMs, via linear probing. The authors also discover VLMs can be trained to generate class labels with a comparable accuracy as their visual encoders. Finally, they analyze the frequency of different classes in the original datasets that were used to train the VLMs, and find that the frequency of samples for a class is positively correlated to the accuracy of the VLMs on that class. This reveals that the poor performance of VLMs in image classification is due to the lack of class labels in their instruction-tuning data.

They also construct a new question-answering dataset, ImageWikiQA, to test the model ability of answering questions related to the finegrained image class. The authors then finetune VLMs using a dataset that consists of both the image classification data and the original nstruction-tuning data. The resulting VLMs can perform well on the ImageWikiQA dataset.

**Strengths:**

- This paper provides a relatively extensive analysis of why SOTA public VLMs perform poorly in image classification. The results generally look correct but there are some issues (as detailed in Weaknesses and Questions) that make them less sound.
- The authors create a new dataset, ImageWikiQA, to evaluate the question-answering capacity of VLMs related to the fine-grained classes in ImageNet.
- The authors show that incorporating the ImageNet classification data into VLM's instruction-tuning data can improve the model performance on ImageWikiQA.

**Weaknesses:**

- It seems that the authors did not consider there may be multiple textual labels for many ImageNet classes. For example, for class with ID n01496331, the class name can be electric ray, crampfish, numbfish, or torpedo. When calculating VLM classification accuracy, the authors can match the model-generated text with any of these class names.
- It may harm the VLM's instruction-following capacity to incorporate the ImageNet 1.28M classification data into the original LLaVA instruction-tuning data to train the VLM, because the classification data has a single fixed template "What type of object is in this photo? ASSISTANT: <Class Name>." Did the authors perform any evaluation of the VLM's instruction-following capacity?
- It is expected that the information necessary for classification is largely preserved in the visual encoders in the VLMs, as these visual encoders are typically pre-trained on ImageNet and are kept frozen during the integration into VLMs. If the instruction-tuning data for VLMs do not contain class-specific information, it is unlikely that the VLMs can automatically align the class information in the visual encoder's output to the text generation. This explains why inference methods do not work in the paper.
- Section 4.2 is a bit unclear. For example, it mentions that the authors randomly sampled at most 3 ImageNet images for each question, but then it mentions there are 2000 multiple-choice questions, each with an image. So how many images are there per question?

**Questions:**

- Can the authors confirm the results in Table 3 (the right sub-table) are the accuracy on the validation/test sets? The accuracy on ImageNet looks very high to me. I have the same question for the accuracy mentioned in line 279.
- In Table 4, LLaVA1.5-7B with GT Class provide still has a relative low accuracy. Why is that?
- Can the authors confirm there are no training data in ImageWikiQA (i.e., it is only a testing dataset)?

**Limitations:**

Please see the Weaknesses.

---

> ### Author Rebuttal · Authors · 2024-08-05
>
> We thank Reviewer 8fB5 for providing detailed and thoughtful feedback on our work. We address Reviewer 8fB5’s questions below.
>
> ---
>
> **Multiple textual labels**
>
> > There may be multiple textual labels for many ImageNet classes. For example, n01496331 can be electric ray, crampfish...
>
> Thank you for your suggestion. **We have now considered ImageNet label synonyms [1] in our evaluation process.** When evaluating with synonyms, **the accuracy only improves by 1%-3%**, which still results in a significant performance gap compared to CLIP.
>
> We will include these results in the revised paper.
>
> [1] https://gist.github.com/yrevar/942d3a0ac09ec9e5eb3a
>
> | Model | Accuracy w/o Synonyms | Accuracy w/ Synonyms | Delta |
> | - | - | -| - |
> | BLIP2-2.7B | 25.3 | 27.8 | 2.5 |
> | IBLIP-7B | 14.6 | 16.5 | 1.9 |
> | IBLIP-13B | 14.7 | 16.6 | 1.9 |
> | LLaVA1.5-7B | 22.8 | 24.6 | 1.8 |
> | LLaVANeXT-V7B | 29.4 | 32.2 | 2.8 |
> | LLaVA1.5-13B | 24.3 | 26.0 | 1.7 |
> | LLaVANeXT-M7B | 32.3 | 35.1 | 2.8 |
> | Claude3 | 53.6 | 56.3 | 2.7 |
> | GeminiPro | 39.2 | 42.5 | 3.3 |
> | GPT4 | 48.5 | 51.1 | 2.6 |
>
> ---
>
> **Instruction-following capacity**
>
> > It may harm the VLM's instruction-following capacity to incorporate the ImageNet 1.28M classification data into the original LLaVA instruction-tuning data.
>
> Great question! **We have now evaluated the original VLM and our fine-tuned VLM on three additional instruction-following benchmarks:** TextVQA, POPE, and MMVet. We found that **fine-tuned VLM achieves the same accuracy as without fine-tuning**.
>
> We will release all the codes and models to reproduce these results and include them in the revised paper.
>
> |  | TextVQA | POPE Popular | POPE Adverse | MMVet |
> | --- | --- | --- | --- | --- |
> | LLaVA1.5-7B (Official Released) | 58.2 | 86.1 | 84.2 | 31.1 |
> | LLaVA1.5-7B (Finetuned) | 58.0 | 86.3 | 84.5 | 31.1 |
>
> ---
>
> **Expected conclusion**
>
> > It is expected that the information necessary for classification is largely preserved in the visual encoders in the VLMs.
>
> We agree that it is apparent that the information necessary for classification is preserved in visual encoders. However, **it is not unclear whether the information still remains after propagating through all the LLM layers** (e.g., 32 layers for Vicuna-7B). Our results highlight that information is preserved after the LLM propagation rather than preserved in visual encoders.
>
> > If the instruction-tuning data for VLMs do not contain class-specific information, it is unlikely that the VLMs can automatically align the class information in the visual encoder's output to the text generation
>
> **This claim is actually unknown and controversial.** The key question is: if both vision encoders and language models have seen a specific class during their uni-modal training, do VLMs need to see the exact class in a multi-modal format to align them during multi-modal training, and how much data is required? **Many previous works have shown that this alignment stage is very data-efficient and even unnecessary [1, 2].**
>
> **Our paper demonstrated that 1) multi-modal data is necessary for alignment, and 2) increasing the data amount leads to linearly improved performance.** This provides a critical data-centric view for VLM training. **Recent work from DeepMind echoes our findings.** They found that most VLM tasks benefit significantly from longer pre-training with more data, as shown in Figure 4 and Appendix K [3].
>
> We will add these clarifications in the revised paper.
>
> [1] Visual Instruction Tuning
>
> [2] Prismatic VLMs: Investigating the Design Space of Visually-Conditioned Language Models
>
> [3] PaliGemma: A versatile 3B VLM for transfer
>
> ---
>
> **ImageWikiQA clarification**
>
> > It mentions that the authors randomly sampled at most 3 images for each question, but then it mentions each with an image. How many images are there per question?
>
> **There is only one image per question** (see Appendix Table 11 for examples).
>
> The confusion arises from our two-stage creation pipeline:
>
> - **1st stage: generate questions at the class level.** For any given class out of the 1000 ImageNet classes, we create questions such as “What is the native region of <class name>?”.
> - **2nd stage: generate questions at the image level.** Each ImageNet class has 50 images, from which we randomly sample 3 images and create 3 questions like “What is the native region of <image i>?”
>
> We will revise the text for clarity.
>
> ---
>
> **Data split clarification**
>
> > Are the results in Table 3 and line 279 are the accuracy on the validation/test sets? The accuracy on ImageNet looks very high to me.
>
> **Both Table 3 and Line 279 are evaluated on the validation set.** The data splits use the official data split (e.g., ImageNet contains 1.28M training images and 50K validation images). Detailed data splits are provided in Appendix Table 6.
>
> **The very high accuracy on ImageNet is one of the significant contributions of our paper.** We found that, after fine-tuning the VLM on ImageNet, there is no more gap between VLM and CLIP, with VLM now being the state-of-the-art classifier.
>
> ---
>
> **Low accuracy of LLaVA1.5-7B with GT class**
>
> > In Table 4, LLaVA1.5-7B with GT Class still has a relative low accuracy. Why?
>
> Table 4 reports the accuracy on ImageWikiQA. **These questions come from Wikipedia and require extensive knowledge, which is very challenging even for humans** (e.g., what is the native region of the guinea pig?)
>
> **The low accuracy (55.9%) is because LLaVA1.5-7B lacks sufficient world knowledge for these ImageNet classes.** It is well known that smaller LMs like Vicuna-7B lack world knowledge compared to larger LMs like GPT4.
>
> ---
>
> **ImageWikiQA test only**
>
> > No training data in ImageWikiQA?
>
> **Yes, ImageWikiQA only contains a test set.** This dataset serves as an important resource to evaluate VLM’s fine-grained classification capabilities as well as its knowledge and reasoning abilities.
>
> ---
>
> Thank you again for your feedback. Please let us know if you have further questions!

---

> > ### Comment · Reviewer_8fB5 · 2024-08-12
> >
> > Thank you for the responses. Most of my concerns are addressed. I have one follow-up question. How many unique questions are there in the ImageWikiQA dataset? For example, a question like “What is the native region of <image i>?” counts only once even though it can be asked for different images and classes.

---

> > > ### Author Response · Authors · 2024-08-12
> > >
> > > Thank you for your reply! We are glad to hear that our response has solved most of your concerns.
> > >
> > > > How many unique questions are there in the ImageWikiQA dataset? For example, does a question like “What is the native region of ?” count only once, even if it’s asked for different images and classes?
> > >
> > > The ImageWikiQA dataset contains 2,000 unique questions. A question like “What is the native region of <image i>?” is counted multiple times if it is asked for different images or classes. For example, if the question is asked for 2 cat images and 2 dog images, it would count as 4 questions in total. To avoid over-representing any particular question, a question like “What is the native region of <image i>?” is limited to a maximum of 3 occurrences across the 2,000 questions.
> > >
> > > Please let us know if you have any further questions!

---

> > > > ### Comment · Reviewer_8fB5 · 2024-08-13
> > > >
> > > > Thanks for the response. I will raise my rating to 5.

---

> > > > > ### Author Response · Authors · 2024-08-13
> > > > >
> > > > > Thank you! We will incorporate these in the revised text and are happy to answer any other concerns you have in mind.

---

> ### Author Response · Authors · 2024-08-09
> **Gentle request for discussion**
>
> Dear Reviewer,
>
> We kindly ask for your feedback on our rebuttal. We have conducted all the requested experiments (ImageNet multiple textual labels and instruction following capacity) and clarified concerns that due to misunderstanding.
>
> If our rebuttal has addressed some of your concerns, we would appreciate it if you could reconsider your scores for our paper. Should you have any further questions, please do not hesitate to reach out to us.
>
> Thank you for your time and consideration!

---

### Official Review · Reviewer_SHnR · 2024-07-12

**Soundness:** 3
**Presentation:** 3
**Contribution:** 3
**Rating:** 7
**Confidence:** 4

**Summary:**

The paper presents an interesting observation of VLMs lagging in image classification performance as compared to the visual encoders lie CLIP used within them. Several hypothesis are explored to explain this observation including train-time (information loss, training objective used), inference-time (prompt variations, label set size) and data related reasons. Their analysis shows the primary reason for the observed gap to be data prevalences, showing a correlation between class prevalence during VLM training and performance in those classes for image classification. The paper then simply proposes inclusion of these datasets into VLM training as a way to fix this issue, showing image classification improvements.

**Strengths:**

1. The paper is nicely structured. An interesting performance trend is observed, hypothesis are explored to explain the phenomena, and conclusions drawn from the experiments are further tested. The evaluation and experimental details, the different hypothesis spanning both training and inference, and the well crafted control experiments make it a good study.
2. The paper is also well motivated and studies a relevant failure mode of VLMs. The results show that even though these modes might encode object or image level concepts, they still suffer from an inability to classify images which is supposed to be a more simpler, and more importantly a more fundamental visual task. This makes it a valuable research problem.
3. The specific experiments to explore each hypothesis are interesting in their own right. For example studying the effects of prompt ordering or CoT to image classification or different variations of inference or label set size, type of objective used, and linear probing results, all provide useful cues towards the hypothesis, but also showcase interesting VLM behavior.
4. The paper proposes a very simple and effective way to bridge the VLM performance gap by fine-tuning on the classification data. It shows how doing so leads to not just regaining high image classification performance, but also improves VLM-specific tasks on the classification dataset, for which they curate a new dataset.

**Weaknesses:**

1. I'm not completely convinced with the paper's final conclusion of data being the reason why CLIP models are superior to VLMs. The fact that linear probes can extract good classification performance from VLMs shows that its a decoding problem since even these models were trained with the same data having skewed prevalences for certain badly performing classes, but the linear probe tuning manages to bring that out to the same degree as CLIP. This implies that either what's missing is a task-aligned fine-tuning (which is what a linear probe does, and which is what the paper does when they fine-tune) or perhaps even a modification of how these are inferred on for image classification which can lead to a more a task-aligned inference (CLIP is trained for text/concept alignment and is evaluated using knn whereas VLMs are trained for vision-text conditioned text generation, but evaluated in a very specific way by attaching tokens of the different options for image classification). In either case, the data argument for explaining the gap might not be the only, or even the strongest reason.
2. The paper proposes inclusion of the image classification datasets as a solution for this problem. If this VLM limitation stems for other reasons such as multimodal confusion and text feature interference or hallucination problem specific to the generated text space, adding more data for different tasks as a solution might not generalize to similar VML issues. In this case, since the new dataset involves tasks which necessitate good image classification, performance after fine-tuning goes up on both sets of tasks. This might not hold true for other equally fundamental visual tasks.

**Questions:**

1. In Figure 2, the label set size does seem to have some effect on the performance gap b/w CLIPs and VLMs. Since this relates to the way VLMs are evaluated for image classification (attaching class options as tokens in the prompt which grow with the label set), do you think this experiment shows the evaluation difference between the two methods (and also how they are originally trained), can explain some of the gap? Also, why do you think the Caltech and ImageNet datasets behave a little differently than Cars and Flowers in Figure 2?
2. Why do you think CLIP does not suffer from the class prevalence bias but LLava does in Figure 3? What happens if we plot this for linear probed LLava models? What happens if we plot this for linear probed Llava models which are fine-tuned on similarly biased data having similarly skewed frequencies? If those models do not show this strong correlation, it might be more about the fine-tuning and less about the data distribution.

**Limitations:**

The paper has discussed limitations.

---

> ### Author Rebuttal · Authors · 2024-08-05
>
> We thank Reviewer SHnR for their positive comments and thoughtful feedback. We address Reviewer SHnR’s questions below.
>
> ---
>
> **Main reason and solution**
>
> > I'm not completely convinced with the paper's final conclusion of data being the reason why CLIP models are superior to VLMs… The data argument for explaining the gap might not be the only, or even the strongest reason.
> >
>
> We clarify that data is not the **primary reason**. The primary reason is that the information required for classification is encoded in the VLM’s latent space but cannot be effectively decoded. Data serves as an **effective solution** to decode this information, as our experiments show that with sufficient data, VLMs can match the accuracy of state-of-the-art classification models.
>
> > This implies that either what's missing is a task-aligned fine-tuning or perhaps even a modification of how these are inferred on for image classification which can lead to a more a task-aligned inference.
>
> Thank you for the thought-provoking question. We agree that task-aligned fine-tuning or inference, such as adding a linear probing loss on the VLM output space for training and using KNN in the VLM output space for evaluation, can improve VLM classification performance.
>
> **However, our focus is on enhancing VLM’s general capabilities in solving a variety of tasks rather than the specific classification task.** The only natural inference interface for different tasks is text generation. Task-aligned inference, like KNN, is only applicable to classification and not other tasks.
>
> For task-aligned fine-tuning, such as adding a new token and a linear probing loss, we observed no improvement when evaluating this fine-tuned VLM on ImageWikiQA. This shows that task-aligned fine-tuning does not improve VLM’s overall capabilities.
>
> In summary, without changing VLM training or inference to maintain its general capabilities for different tasks, adding data is the most promising and effective approach. We will add these clarifications in the revised paper.
>
> ---
>
> **Solution to other problems**
>
> > If this VLM limitation stems for other reasons such as multimodal confusion and text feature interference or hallucination problem specific to the generated text space, adding more data for different tasks as a solution might not generalize to similar VML issues.
>
> Thank you for your question. We agree that for issues like multimodal confusion or hallucination, adding data might not resolve the problem, but **addressing these is beyond the scope of our paper.**
>
> The main contribution of our paper is the formulation of the problem that VLMs are poor image classifiers and a thorough investigation into why this is the case and how to solve it. Our paper demonstrates that the poor classification performance of VLMs is due to the inability to decode the information encoded in VLMs. **Given this reason, data serves as an effective solution to decode the information.**
>
> ---
>
> **Label set size analysis**
>
> > Do you think the label set size experiment shows the evaluation difference between CLIPs and VLMs can explain some of the gap?
>
> Thank you for your question. **The label set size experiment can partially explain the gap but not entirely.**
>
> By reducing the number of labels for classification, we can narrow the gap between VLM and CLIP, but a gap remains across all label set sizes, even with just two labels (two-way classification).
>
> Moreover, we find that while the absolute gap between VLMs and CLIPs narrows with reduced label size, the relative gap increases (Appendix Figure 4). For example, in two-way classification on ImageNet, VLMs have a 5.7% error rate, while CLIP has a 0.2% error rate, resulting in a 28.5x gap; for 20 classes, VLMs have an 18.0% error rate, while CLIP has a 2.5% error rate, resulting in a 7.2x gap.
>
> **These results indicate that the VLM-CLIP gap cannot be fully explained by label set size.**
>
> > Why do you think the Caltech and ImageNet datasets behave a little differently than Cars and Flowers in Figure 2?
>
> The absolute gap on Flowers and Cars is larger than ImageNet and Caltech when reducing the label set size. **This may be because Flowers and Cars are more fine-grained classification datasets, while ImageNet and Caltech are more coarse-grained**. VLMs are weaker in fine-grained classification compared to CLIP, resulting in a larger gap.
>
> We will add these clarifications in the revised paper.
>
> ---
>
> **Class prevalence bias**
>
> > Why do you think CLIP does not suffer from the class prevalence bias but LLava does in Figure 3?
>
> Great question! **In Figure 3, the class frequency on the x-axis is computed based on the LLaVA pre-training and instruction-tuning dataset, not the CLIP training dataset.** If we change the x-axis to the CLIP pre-training data frequency, CLIP should also show prevalence bias. Please refer to Figure 2 in [1].
>
> [1] No “Zero-Shot” Without Exponential Data: Pretraining Concept Frequency Determines Multimodal Model Performance.
>
> > What happens if we plot this for linear probed LLava models?
>
> **We plotted the line for the linear probed LLaVA model, which aligns with the fine-tuned LLaVA (see PDF in general response).** The zero correlation is expected because fine-tuning or probing on the ImageNet dataset equalizes class frequencies, as ImageNet classes are evenly distributed. We will include this line in the updated paper.
>
> > What happens if we plot this for linear probed Llava models which are fine-tuned on similarly biased data having similarly skewed frequencies?
>
> We have not yet trained a new model on skewed data, but **we found that training on the ImageNet classification dataset does not improve performance on the Flowers dataset, which has drastically different class frequencies.** This further verifies that data, rather than fine-tuning, is the key factor determining performance. We will add this in the revised paper.
>
> ---
>
> Thank you again for your feedback. Please let us know if you have further questions!

---

> > ### Comment · Reviewer_SHnR · 2024-08-14
> >
> > Thanks you for the elaborate discussion and answering all my questions related to label set size and class prevalences. The new experiments you added were very helpful.
> >
> > Some responses:
> > > We clarify that data is not the primary reason. The primary reason is that the information required for classification is encoded in the VLM’s latent space but cannot be effectively decoded. Data serves as an effective solution to decode this information, as our experiments show that with sufficient data, VLMs can match the accuracy of state-of-the-art classification models.
> >
> > Thanks for clarifying this! The paper mentions it in multiple places that data is the reason why VLMs are bad at classification. Here's one of the prominent concluding statement at the end of Section 3: *"These results suggest that data is the primary cause of the poor classification performance of VLMs."*
> > If the main reason is the inability to decode the latent information, and the solution to it is to fine-tune with classification data, then I would advise the authors to explicitly state that in the introduction to prevent confusion. Currently Figure 1 simply says "Not Enough Data" as the reason which is not true. It's more about being able to align the VLM features using the right kind of data.
> >
> > The paper successfully shows that VLMs hold the information needed for performing well at image classification (so do their CLIP-based visual encoders), but require some kind of data-aligned fine-tuning on classification or instruction tuning to do well on those tasks. This kind of tuning being necessary is not surprising since you need some way to align the VLM to classes seen at classification time. It's a known observation and often seen in other models as well, but the paper conducts hypothesis-driven experimentation to rule out other possibilities, and in the process generates new ablative results and a new dataset, which is valuable to the community. Illuminating the general problem of VLM needing alignment for classification is a useful direction for future work. I will increase my rating.

---

> > > ### Author Response · Authors · 2024-08-14
> > >
> > > We are pleased to hear that our response has addressed your concerns, and we will certainly revise the related text as you suggested. Thank you for your time and consideration!

---

> ### Author Response · Authors · 2024-08-09
> **Gentle request for discussion**
>
> Dear Reviewer,
>
> We kindly request your consideration of our rebuttal. We believe we have thoroughly explained why the data presents a promising solution rather than the root of the problem, and why altering the training or inference objective would not be appropriate. Additionally, we have addressed the other questions raised.
>
> Thank you for your time and consideration!

---

> ### Author Response · Authors · 2024-08-13
>
> Dear Reviewer SHnR,
>
> With just 1 day left in the response period, we would greatly appreciate it if you could kindly review our responses soon. We are still looking to discuss any remaining concerns.
>
> Thank you for your time!

---

### Official Review · Reviewer_fPsb · 2024-07-12

**Soundness:** 2
**Presentation:** 2
**Contribution:** 2
**Rating:** 5
**Confidence:** 3

**Summary:**

In this paper, the authors explored why Vision-Language Models (VLMs) significantly underperform as image classifiers. They compared several publicly available with proprietary VLMs on several classification benchmark datasets, including ImageNet, Flowers102, StanfordCars, Caltech101, and their newly collected ImageWikiQA dataset. The paper presented multiple hypotheses regarding the underperformance of VLMs, addressing questions related to inference, training objectives, and training/finetuning data. The authors conducted empirical studies and concluded that the performance of VLMs in classification is determined by the data used. They fine-tuned LLaVa1.5-7B on ImageNet and the LLaVa instruction-tuning dataset to enhance classification accuracy.

**Strengths:**

1. The paper is well-organized and easy to follow.
2. The motivation is clear, and the empirical study is thorough.
3. The authors analyzed the training data of LLaVA1.5 models to show the strong correlation between class frequency over accuracy.

**Weaknesses:**

1. The authors did not provide much analysis on the proprietary VLMs. Can the authors address the hypotheses related to inference, training objectives, and data for these VLMs?
2. The authors introduced “open-world setting” and “closed-wold setting” as evaluation protocols in section 2.3. However, the protocol seems to be only applied to Table 1. It is not clear which settings are being used for sections 3 and 4.
3. The authors need to provide more explanation for Table 2. What is the difference between the “Closed-World Setting” performance of Table 1 and the “Base Prompt w/ Label (Fixed Order)” in Table 2? Why are their accuracies different?
4. The LLaVA-7B and BLIP2-7B models are not large enough to demonstrate intrinsic capabilities such as chain-of-thought reasoning [1]. The authors could include the prompt variation results of Table 2 for one of the proprietary VLMs to show the impact on performance.
5. In Lines 178-179, the authors observed that the information necessary for classification is preserved in VLMs' latent space; however, it cannot be effectively decoded. If the VLMs have the essential information, then why are they not able to decode it? The authors should explain the possible reasons for this.
6. In Table 4, the finetuned VLM performance is worse than the zero-shot performance of GPT4  on ImageWikiQA. What are the possible reasons for this? Why is the “Finetuned on ImageNet” model performance (30.6) worse than the non-finetuned model performance (37.8)? Does finetuning always enhance the general capabilities of the VLMs?

[1] Zhang, Zhuosheng, Aston Zhang, Mu Li, Hai Zhao, George Karypis, and Alex Smola. "Multimodal chain-of-thought reasoning in language models." arXiv preprint arXiv:2302.00923 (2023).

**Questions:**

Please respond to the points of weakness I mentioned above.

**Limitations:**

Yes.

---

> ### Author Rebuttal · Authors · 2024-08-05
>
> We thank Reviewer fPsb for their positive comments and thoughtful feedback. We address Reviewer fPsb’s questions below.
>
> ---
>
> **Analysis of proprietary VLMs**
>
> > Can the authors address the hypotheses related to inference, training objectives, and data for proprietary VLMs?
>
> Thank you for your question. **Proprietary VLMs haven’t released the training details and data usage, so it is impossible to directly address training-related or data-related hypotheses.** For inference-related hypotheses, we conduct additional experiments with GPT4 (**see PDF in general response**), and the conclusion is the same as the public VLMs.
>
> **Recent work from DeepMind** [1] shows that most VLM tasks benefit significantly from longer pre-training with more data (in Figure 4 and Appendix K of [1]). This **aligns with our main conclusion that data is the critical factor in improving VLM performance.**
>
> [1] PaliGemma: A versatile 3B VLM for transfer.
>
> ---
>
> **Open-world vs closed-world setting**
>
> > It is not clear “open-world” vs “closed-world” settings are being used for sections 3 and 4.
>
> Great question! **Sec. 3 and 4 all use “open-world” settings (not providing the label set in the prompt), except for prompt variation analysis and label set size analysis in Sec. 3.1, using “closed-world” settings (providing the label set in the prompt).**
>
> Naturally, the “closed-world” setting narrows the model’s generation space and increases accuracy. When the model can easily predict the class in the label set by modifying the inference space (probabilistic inference in Sec. 3.1 and probing VLM in Sec. 3.2) or training VLMs with more classification data (Sec. 4), **the advantage of the “closed-world” setting doesn’t exist**. Thus, for these experiments, we use the “open-world” setting.
>
> We will clarify this in the revised paper.
>
> ---
>
> **Clarification for Table 2**
>
> > What is the difference between the “Closed-World Setting” performance of Table 1 and the “Base Prompt w/ Label (Fixed Order)” in Table 2? Why are they different?
>
> **Table 1’s “Closed-World Setting” is equivalent to Table 2’s “Base Prompt w/ Label (Random Order)”.** In this setting, we concatenate label candidates in a random order for each question: “What’s in the image? Choose one from random_shuffle([cat, dog, pig])”. The accuracy for these two settings is identical between Table 1 and Table 2.
>
> Table 2’s “Base Prompt w/ Label (Fixed Order)” concatenates label candidates in a fixed order for each question: “What’s in the image? Choose one from [cat, dog, pig]”. This setting is used to rule out the possibility that the order of labels might affect model performance.
>
> ---
>
> **CoT with larger VLMs**
>
> > The LLaVA-7B and BLIP2-7B models are not large enough to demonstrate intrinsic capabilities such as chain-of-thought reasoning [1].
>
> Thank you for your suggestion! We have now evaluated GPT-4’s performance on ImageNet using chain-of-thought (CoT) reasoning.
>
> In Table 2, GPT-4 achieves 60.6% accuracy on ImageNet without CoT. **With CoT, GPT-4 achieves 62.2% accuracy**, demonstrating that CoT has a limited impact on image classification, even for large VLMs.
>
> We will add these results and citations in the revised paper.
>
> ---
>
> **Cannot decode information**
>
> > If the VLMs have the essential information, then why are they not able to decode it?
>
>
> Great question! **One possible reason is that the VLM’s decoding space is too large and not aligned with the visual features.** Specifically, VLM decoding is performed through next-word prediction, which usually involves a vocabulary of over 10,000 words. The output text embedding is not aligned with the visual features due to insufficient data to align these spaces.
>
> **In general, having information in a model does not necessarily mean the model can express that information.** This phenomenon is also observed in other research areas. For example, after training ResNets or ViTs with self-supervised learning methods like SimCLR, the model acquires discriminative information for different classes. However, this information can only be decoded by adding and training a linear layer on a new dataset.
>
> Similarly, we demonstrate that VLMs possess classification information, but it is not readily expressible. **Adding classification-related data helps bridge the gap between “information possession” and “information expression”** (refer to Table 3 in our paper).
>
> We will include this discussion in the revised paper.
>
> ---
>
> **Fine-tuning results**
>
> > In Table 4, why is the “Finetuned on ImageNet” model performance (30.6) worse than the non-finetuned model performance (37.8 [should be 38.0])? Does finetuning always enhance the general capabilities of the VLMs?
>
>
> **Fine-tuning often improves in-distribution performance but does not necessarily enhance out-of-distribution or general capabilities.**
>
> **Table 4 presents results of models fine-tuned on ImageNet and evaluated on ImageWikiQA, which are vastly different datasets.** ImageNet questions only require classification, such as “Q:  <image> What is in the image? A: dog,” while ImageWikiQA questions demand both classification and knowledge/reasoning, such as “Q:  <image> What is the native region of this object? A: South America.”
>
> Fine-tuning solely on ImageNet trains the model to classify, but it can lead to a loss of general capabilities like reasoning and knowledge, resulting in lower performance on ImageWikiQA (30.6 fine-tuned vs. 38.0 pre-trained). This phenomenon is known as “catastrophic forgetting” [1].
>
> **However, when we fine-tune on both ImageNet and instruction tuning data, the model learns classification while retaining its original capabilities, leading to a significant performance improvement on ImageWikiQA (49.8 fine-tuned vs. 38.0 pre-trained).**
>
> We discussed this in Lines 281-285 and will add further clarification.
>
> [1] Overcoming catastrophic forgetting in neural networks.
>
> ---
>
> Thank you again for your feedback. Please let us know if you have further questions!

---

> ### Author Response · Authors · 2024-08-09
> **Gentle request for discussion**
>
> Dear reviewer,
>
> We would kindly ask for a response to our rebuttal. We believe that some of your concerns are due to a misunderstanding and are not weaknesses, and we would appreciate it if you would revisit your evaluation of our work. Thank you for your time and consideration!

---

> ### Author Response · Authors · 2024-08-13
>
> Dear Reviewer fPsb,
>
> With just 1 day left in the response period, we would greatly appreciate it if you could kindly review our responses soon. We are still looking to discuss any remaining concerns.
>
> Thank you for your time!

---

### Author Rebuttal · Authors · 2024-08-05

We thank the reviewers for their thoughtful feedback on our manuscript. Below, we provide individual responses to each reviewer. Please let us know if you have any further questions or concerns!

**We have also attached a PDF for reviewers fPsb and SHnR, which includes visualizations to further address their questions.**

---

### Author Response · Authors · 2024-08-07
**Please review our response**

Dear Reviewers,

We appreciate your valuable feedback and have provided detailed responses to all your comments. We kindly ask you to review our responses. Should you have any further questions, please do not hesitate to initiate a discussion with us.

Thank you for your time and consideration!

---

### Decision · Program_Chairs · 2024-09-25

**Decision:**

Accept (poster)

**Comment:**

The work reveals a problem that vision language models (e.g. GPT-4o) including proprietary and open-source ones perform poorly in image classification (e.g. ImageNet) compared to CLIP while they are both trained on the Internet data. The question is at which layer do VLMs contain necessary discriminative features? And they find the answer is the last two layers of VLMs i.e., finetuning a linear prob on VLMs improve the classification accuracy. The experiments are quite extensive: 7 open-source VLMs, 3, proprietary VLMs, 4 different datasets.

Initially, the reviews were mixed (7, 5, 5, 3). After the rebuttal, no reviewers express any concerns. The AC agrees that the authors have addressed most of the concerns raised and recommends `Accept`.